# Dynamic subcellular proteomics identifies regulators of adipocyte insulin action

Olivia J. Conway[1], Josie A. Christopher [2], Lisa M. Breckels [2], Hanqi Li[3], Dilip Menon[1], Meghna Birla[1], Bethan L. Hawkins [1], Lu Liang[1], Satish Patel [1], Francoise Koumanov [4], David C. Gershlick [3], David B. Savage[1], Michael P. Weekes [3,5], Kathryn S. Lilley [2] & Daniel J. Fazakerley [1] ✉

Insulin acts on adipocytes to suppress lipolysis and increase glucose uptake to control whole-body glucose and lipid metabolism. Regulation of these processes by insulin signalling depends on changes in protein localisation. However, the extent of insulin-stimulated changes to the adipocyte spatial proteome, and the importance of these in the cellular insulin response, is unknown. Here, we use subcellular proteomics approaches to map acute insulin-stimulated protein relocalisation in adipocytes on a cell-wide scale. These data reveal extensive insulin-regulated protein redistribution, with hundreds of insulin-responsive proteins. These include the uncharacterised protein C3ORF18, which redistributes to the plasma membrane in response to insulin. Studies in C3ORF18-depleted adipocytes suggest this protein is required to maintain adipocyte insulin sensitivity. Overall, our data highlight the scale of protein relocalisation in the adipocyte insulin response, and provide an accessible resource to inform further studies into how changes in protein localisation contribute to cellular insulin responses.

White adipose tissue is sensitive to hormonal and metabolic inputs that balance energy storage and energy release, coordinating adipocyte metabolism and organismal needs. Postprandially, increased circulating insulin promotes adipocyte energy storage by increasing glucose uptake and suppressing lipolysis. This metabolic response to insulin is mediated by an extensive signalling network[1], which elicits cellular responses that require changes in protein subcellular localisation.

Insulin binding to its receptor, INSR, alters the subcellular distribution of specific proteins via several mechanisms. These include modulating protein-protein interactions via protein phosphorylation, recruiting proteins to membranes through lipid messenger generation (e.g., PIE(3,4,5)P3), and through direct effects on membrane traffic. For example, recruitment of insulin receptor substrate 1 (IRS-1) and phosphoinositide 3-kinase (PI3K) to the activated INSR is mediated by

binding to phospho-tyrosine residues on INSR and IRS1, respectively, whilst insulin-stimulated phosphorylation of the tuberous sclerosis complex (TSC)1/2 complex promotes its dissociation from the lysosomes[2]. Insulin signalling also directly targets known regulators of membrane trafficking (e.g., TBC1D4) to deliver glucose transporters (i.e., GLUT4) to the plasma membrane (PM) and promote glucose uptake[3]. Endosomal cargoes, including transferrin receptor (TfR)[4,5], insulin-like growth factor 2 receptor (IGF2R)/cation-independent mannose 6-phosphate receptor (CI-M6PR)[6,7], and sodium/hydrogen exchanger 6 (NHE6)[8], also undergo insulin-stimulated trafficking to the PM, although the physiological significance of endosomal cargo redistribution to the cell surface remains unclear. Collectively, these examples highlight how multiple mechanisms governing protein relocalisation play an essential role in the cellular insulin response. Moreover, insulin-stimulated protein redistribution is impaired in

[1]Metabolic Research Laboratories, Institute of Metabolic Science, University of Cambridge, Cambridge, UK. [2]Cambridge Centre for Proteomics, Department of Biochemistry, University of Cambridge, Cambridge, UK. [3]Cambridge Institute for Medical Research, University of Cambridge, Cambridge, UK. [4]Centre for Nutrition, Exercise and Metabolism, Department for Health, University of Bath, Bath, UK. [5]Department of Medicine, University of Cambridge, Hills Road, Cambridge, UK. ✉e-mail: djf72@cam.ac.uk

states of insulin resistance (e.g., GLUT4 delivery to the PM[9]), exemplifying the importance of regulating protein localisation in metabolic health.

Unbiased approaches to study the acute adipocyte insulin response have, to date, focused solely on protein phosphorylation[1,10], while studies into insulin-regulated protein movement have been limited to studies on single proteins or identifying translocation to/ from a single organelle[8,11]. Here, we combine unbiased global (Localisation of Organelle Proteins by Isotope Tagging after Differential ultraCentrifugation (LOPIT-DC)[12]) and targeted PM profiling[13] subcellular proteomics techniques to map the spatial proteome of 3T3-L1 adipocytes and identify changes in protein localisation in response to an acute insulin stimulus. These studies reveal >500 high-confidence insulin-responsive proteins. To demonstrate the utility of identifying proteins that move in response to insulin, we focus on C3ORF18, which redistributes to the PM following insulin stimulation. Studies in adipocytes depleted of C3ORF18 show that C3ORF18 plays a role in maintaining adipocyte insulin responsiveness. Overall, our work provides a key resource highlighting the dynamic adipocyte subcellular proteome and impetus for future mechanistic studies into how insulin regulates adipocyte function.

## Results

### Mapping the subcellular 3T3-L1 adipocyte proteome using LOPIT-DC

To map the 3T3-L1 adipocyte proteome, we modified the LOPIT-DC subcellular fractionation workflow[12] to include an additional lipid droplet-enriched fraction (Fig. 1a). Cells were lysed and separated into 9 fractions by differential centrifugation, including a cytosol-enriched soluble fraction. To isolate lipid droplets for the tenth fraction, cells were lysed by nitrogen cavitation and floated through a sucrose gradient. Mass spectrometry analysis of the 10 fractions identified 4269 proteins in non-stimulated (basal) replicates ($n = 3$), and 4384 proteins in insulin-stimulated replicates (100 nM, 20 min; $n = 3$), with 4005 proteins identified across all replicates of both conditions (Fig. 1b). Data were processed as previously described[12,14]. A set of organelle marker proteins based on existing curated markers, proteins well established to reside in specific subcellular niches, was used to annotate datasets[12,15,16] (total 531; Supplementary Data 1), and proteins were excluded as marker proteins if they appeared to relocalise in response to insulin. QSep[17], which quantifies the separation of clusters in subcellular proteomics data, was used to confirm the separation of organelles in our annotated data (Supplementary Fig. 1a). The unique distribution of organelles across fractions was further confirmed visually through the profiles of the marker proteins (Fig. 1c, Supplementary Fig. 1b), and the formation of distinct clusters when visualised using linear dimensionality reduction (PCA) (Fig. 1d). In addition, the lipid droplet fraction was enriched for known lipid droplet-localised proteins including the perilipins (PLIN1, PLIN2, PLIN3, PLIN4), VPS13C, CIDEC, galectin-12, and ABHD5 (Supplementary Fig. 1c, d).

Protein localisation under each condition was predicted using the semi-supervised Bayesian machine learning algorithm Bayesian ANalysis of Differential Localisation Experiments (BANDLE)[18]. Accurate localisation assignment requires organelles to have ≥10 marker proteins in all replicates with a distinct distribution across fractions as training data[18,19]. These criteria excluded the use of both lipid droplets and endosomes as organelles for protein allocation using BANDLE. Of the 4005 proteins, 1755 proteins in the basal dataset and 1614 in the insulin-stimulated data were assigned to a single subcellular location with high confidence (allocation probability >0.9; outlier probability <mean outlier probability for that organelle), in addition to the designated marker proteins (Fig. 1e, Supplementary Data 1). Of the allocated proteins, 1367 shared the same localisation under basal and insulin-stimulated conditions. The remaining ~50% of proteins not allocated to an organelle were classified as 'undefined'. A majority of

these are likely multi-localised proteins found in more than one subcellular domain, as previously reported[8]. Protein localisations in basal and insulin-stimulated adipocytes can be viewed interactively at https://proteome.shinyapps.io/adipocyte2025/.

Gene ontology cellular compartment (GoCC) enrichment analysis of proteins allocated to specific subcellular compartments showed enrichment of expected terms (Supplementary Data 1). Comparing our protein organelle assignments to localisations in cultured human adipocytes[20] (Simpson−Golabi−Behmel syndrome (SGBS) adipocytes; Supplementary Data 1) revealed high concordance, particularly for proteins allocated to organelles typically well resolved by centrifugation approaches (i.e., mitochondria, cytosol) (>80%) (Supplementary Fig. 1e). Organelles that are harder to resolve, such as those comprising the secretory pathway (endoplasmic reticulum (ER)-Golgi-PM), had lower concordance, likely reflecting dynamic protein traffic and multilocalisation within this pathway. Accordingly, most proteins localised to the ER, Golgi or PM in 3T3-L1 adipocytes were assigned to organelles within the secretory pathway in SGBS adipocytes (Supplementary Fig. 1e).

Specific analysis of the lipid droplet fraction (fraction 10) revealed enrichment of all septin proteins identified (septin 2, 5, 6, 7, 8, 9, 10, 11) (Supplementary Fig. 1f) and these GTP−binding cytoskeletal proteins clustered near to lipid droplet marker proteins when visualised by PCA (Supplementary Fig. 1g). Septin 9 has been implicated in lipid droplet formation and localisation in non-adipocytes[21,22], whilst septins 7 and 11 are reported to regulate adipocyte lipid storage and metabolism[23–25]. Immunostaining of septin 2 and 11 in mouse and human adipocytes revealed puncta close to lipid droplets in addition to more dispersed staining (Fig. 1f, Supplementary Fig. 1h). Further, knockdown of *Septin2* during differentiation impaired lipid accumulation in the absence of changes in gene expression markers of differentiation (Fig. 1g, Supplementary Fig. 1i–k). Along with significant correlations between adipose tissue septin protein abundance and adipose-related clinical traits (Supplementary Fig. 1l)[26–47], our data support the emerging role for this family of proteins in lipid handling/metabolism in adipocytes. Overall, we provide a high-resolution map of the 3T3-L1 adipocyte proteome, which has high concordance with cultured human adipocytes and use these data to identify that septins localise to lipid droplets in mature adipocytes.

### Insulin stimulates widespread protein relocalisation

Having mapped the adipocyte proteome under basal and acute insulin-stimulated conditions, we next identified proteins with altered localisation in response to insulin by performing differential localisation analysis using BANDLE[18,48]. This Bayesian analysis framework compares the distribution profiles of proteins in unstimulated (basal) and insulin-stimulated adipocytes, and is independent of whether a protein has been allocated to an organelle (as in Fig. 1e). We identified 899 out of 4005 proteins as candidate movers (differential localisation probability (*diff. loc. prob.*) > 0), of which 502 were high-confidence (*diff. loc. prob.* = 1) (Fig. 2a; Supplementary Data 1). Known insulin-responsive proteins, including GLUT4[49] and LNPEP/IRAP[50,51], were predicted to move with high confidence (*diff. loc. prob.* = 1, Fig. 2b), validating our approach to detect insulin-responsive protein translocation. Changes in protein localisation were independent of IRAP and GLUT4 protein abundance (Supplementary Fig. 2a), and we have previously reported that total protein abundance is unchanged across the proteome in response to acute insulin stimulation[10].

Differentially localised proteins were enriched for signalling terms (Fig. 2c), including insulin signalling, and for proteins previously identified to contain insulin-regulated phosphosites in 3T3-L1 adipocytes (Fig. 2d, Supplementary Data 1). The known insulin-regulated phosphoproteins IRS-1 and 2 were predicted to move with high confidence (*diff. loc. prob.* = 1, Supplementary Fig. 2b), whilst several protein kinases, including MARK2, SIK2, and mTOR, relocalised and were

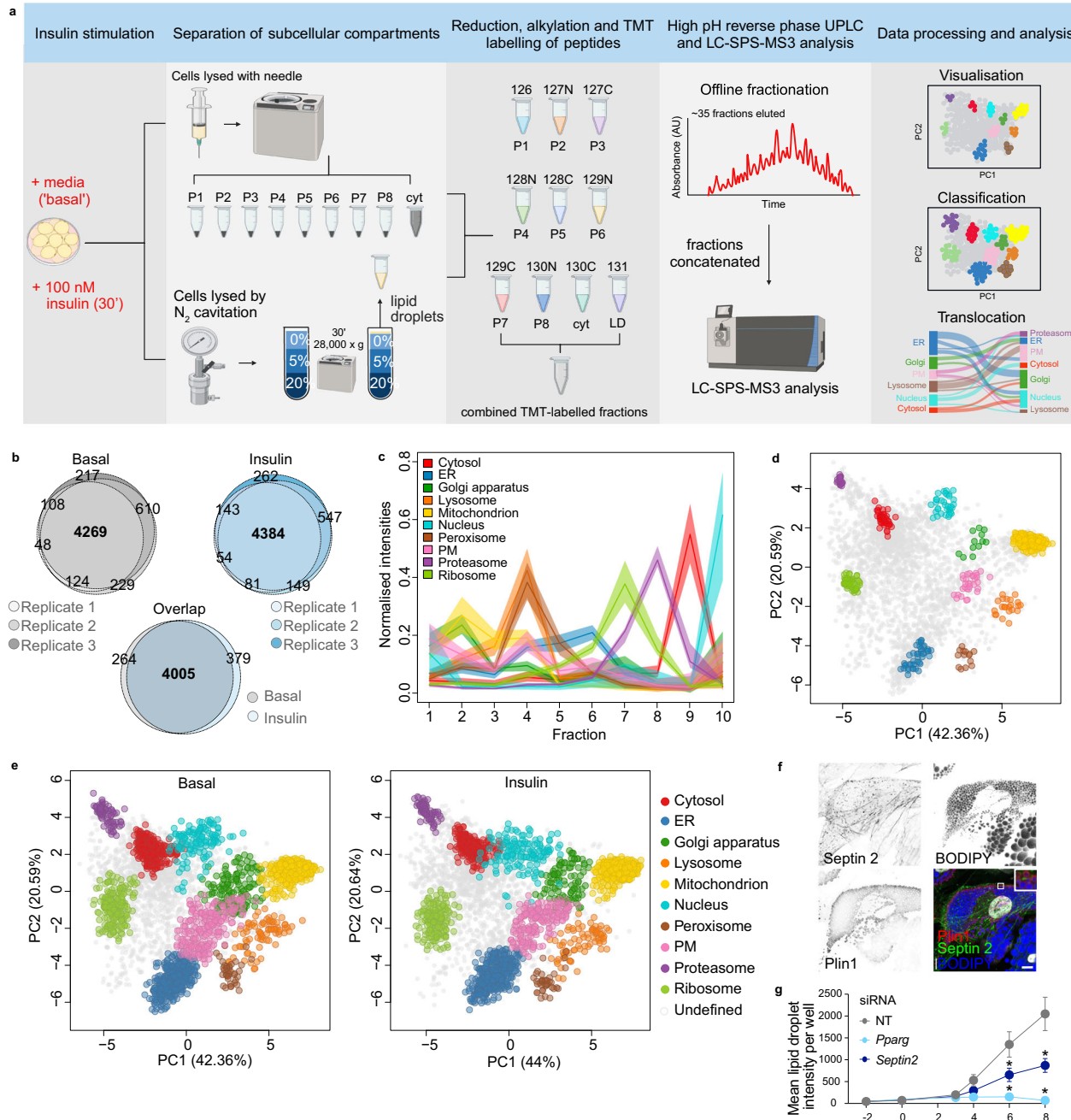

**Fig. 1 | LOPIT-DC maps the spatial proteome of 3T3-L1 adipocytes under basal and insulin-stimulated conditions. a** Schematic of the experimental workflow used. Created in BioRender. Conway, O. (2026) BioRender.com/z3wvppa. **b** Venn diagrams of overlap in proteins identified across three replicates of LOPIT-DC in basal and insulin-stimulated conditions. Data for each condition was concatenated to give 4005 proteins in common across both conditions. **c** Profile of organelle marker proteins across all 10 fractions (basal replicate 1 data shown as mean ± SEM, all replicates in Supplementary Fig. 1b; organelles coloured as red = cytosol, dark blue = endoplasmic reticulum, dark green = Golgi apparatus, orange = lysosome, yellow = mitochondrion, light blue = nucleus, brown = peroxisome, pink = plasma membrane, purple = proteasome, green = ribosome, grey = undefined). **d** Principal component analysis (PCA) projection of concatenated basal data ($n = 3$ biological

replicates) showing distinct clustering of organelle marker proteins. Each point represents an individual protein. **e** PCA projection of proteins allocated to each organelle under basal and insulin-stimulated conditions. **f** Immunostaining of Septin 2, PLIN1 and BODIPY™ 493/503 in SGBS human adipocytes (scale bar = 10 μm). Data are representative of 3 independent experiments. **g** Mean lipid droplet intensity per well in 3T3-L1 adipocytes following siRNA-mediated depletion of *Pparg* (light blue) or *Septin2* (dark blue) on days −2, 0, 3 and 6 of differentiation ($n = 5$ biological replicates for NT and *Pparg*; $n = 4$ for *Septin2*; *Pparg* vs. NT day 6 $p = 0.015$, day 8 $p = 0.014$; *Septin2* vs. NT day 6 $p = 0.042$, day 8 $p = 0.036$), data represented as mean ± SEM; *$p < 0.05$ by two-way ANOVA corrected for multiple comparisons. Source data are provided as a Source Data File.

phosphorylated in response to insulin (*diff. loc. prob.* = 1, 1, 0.14, Supplementary Fig. 2c). In addition, several regulators of mTOR complex 1 (mTORC1) activity were also differentially localised and targeted by insulin signalling. These included the well-characterised target of

insulin signalling, TSC1/2 (*diff. loc. prob.* = 1, Supplementary Fig. 2d), as well as members of the GATOR1 (DEPDC5), GATOR2 (WDR59) and KICSTOR (SZT2) complexes which are typically associated with the amino acid sensing branch of mTORC1 regulation (*diff. loc. prob.* =

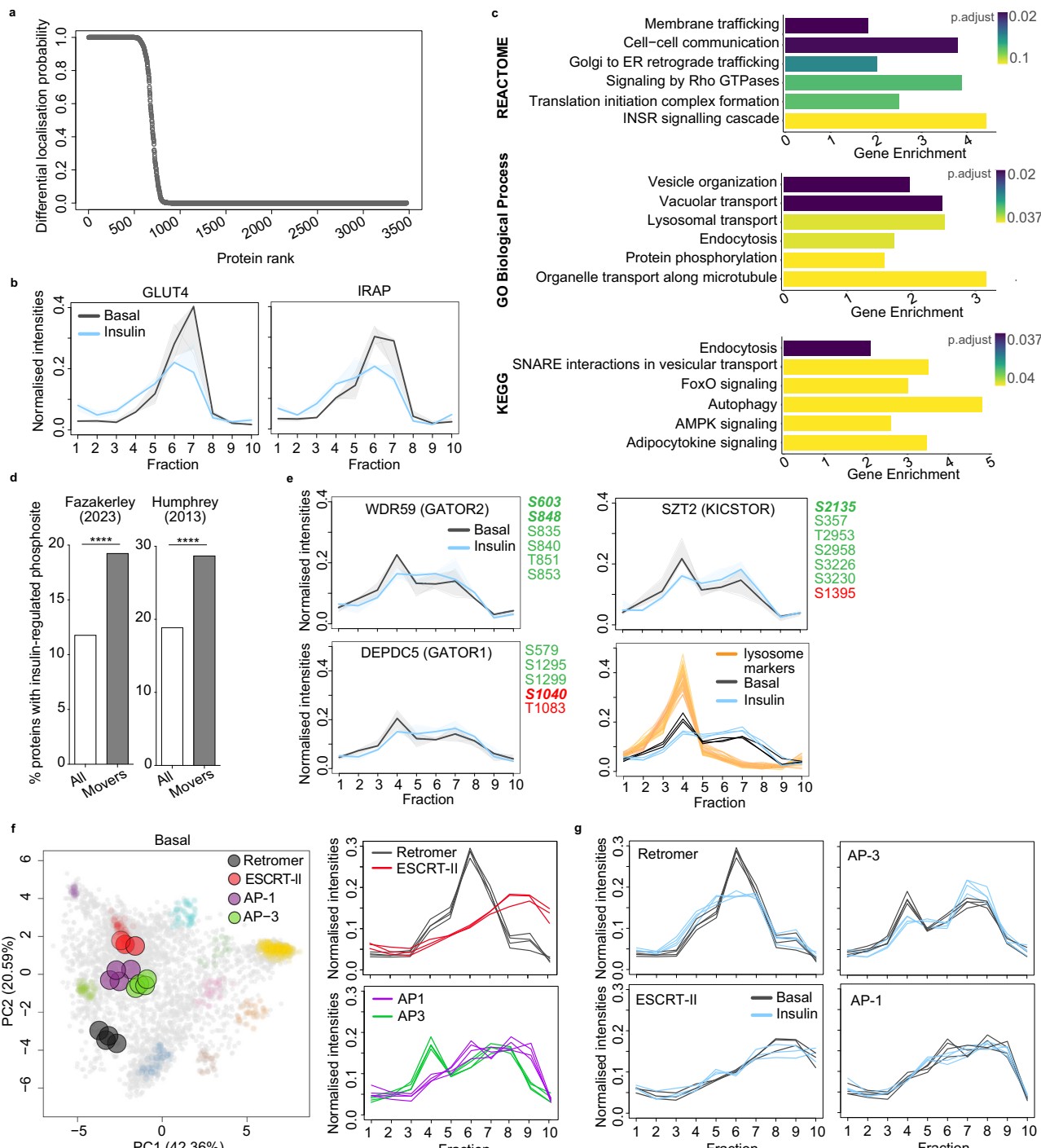

**Fig. 2 | Insulin stimulates extensive changes in protein localisation, including proteins involved in cellular signalling and membrane trafficking. a** Differential localisation probability of all proteins (not including marker proteins) as determined using BANDLE. **b** Distribution of GLUT4 and IRAP across fractions under basal (grey) and insulin-treated (blue) conditions (solid line = mean, shaded area 95% confidence interval (CI), $n = 3$ biological replicates per condition). **c** Gene set enrichment analysis of high confidence differentially localised proteins (*diff. loc. prob.* = 1) using Gene Ontology Biological Process, Reactome and KEGG databases (top 6 terms shown, *p*-values adjusted using the Benjamini-Hochberg method). **d** Enrichment of proteins with insulin-regulated phosphorylation sites in 3T3-L1 adipocytes as reported by Fazakerley (2023)[10] or Humphrey (2013)[1] in all proteins vs. high-confidence differentially localised proteins (*diff. loc. prob.* = 1). Two-sided Fisher's exact test, Fazakerley $p = 2.2 \times 10^{-11}$, Humphrey $p = 1.3 \times 10^{-13}$. **e** Distribution of DEPDC5 (GATOR1), WDR59 (GATOR2) and SZT2 (KICSTOR) subunits across fractions under basal (grey) and insulin-treated (blue) adipocytes (solid line =

mean, shaded area 95% CI, $n = 3$ biological replicates per condition), and insulin-regulated phosphorylation sites reported in Fazakerley (2023)[10] and Humphrey (2013)[1] (green = increased and red = decreased phosphorylation, regular font = found in either Fazakerley and Humphrey, bold italic = found in both Fazakerley and Humphrey). Bottom right panel shows WDR59, SZT2 and DEPDC5 profiles under basal (grey) and insulin-treated (blue) conditions with lysosomal protein markers in orange. **f** PCA projection of concatenated basal LOPIT-DC data with retromer (black; VPS26A, VPS26B, VPS29, VPS35), ESCRT-II (red; VPS25, VPS36, SNF8), AP-1 (purple; AP1B1, AP1G1, AP1M1, AP1S1) and AP-3 (green; AP3B1, AP3D1, AP3M1, AP3S1) complex subunits highlighted (left panel), and mean distribution of subunits across fractions under basal conditions (right panel). **g** Mean distribution of retromer, ESCRT-II, AP1 and AP3 subunits across fractions under basal (grey) and insulin-treated (blue) conditions ($n = 3$ biological replicates per condition). Source data are provided as a Source Data File.

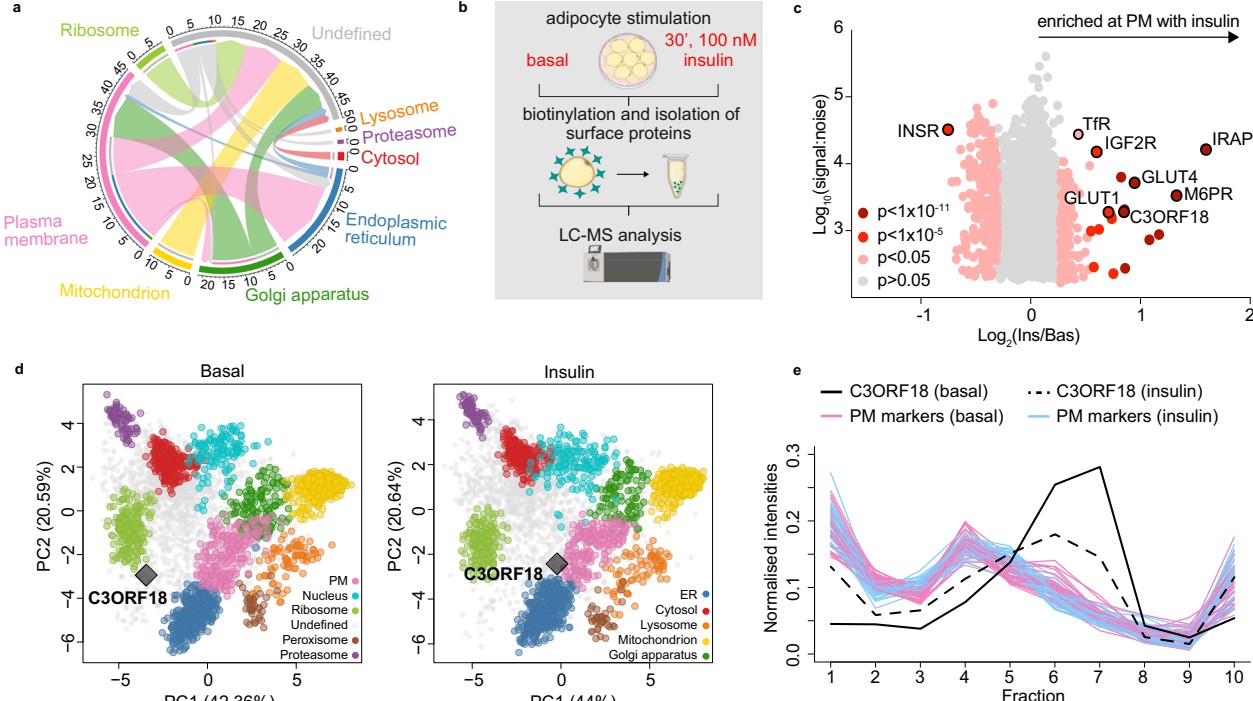

**Fig. 3 | Quantitative plasma membrane proteomics reveals the extent of plasma membrane remodelling in response to insulin. a** Chord plot visualising movement of proteins predicted to relocalise with high-confidence (*diff. loc. prob.* = 1) assigned to an organelle in at least 1 condition. The coloured lines indicate movement between compartments, with the arrowhead indicating the direction of movement. For example, the two dark green blocks emanating from the "Golgi Apparatus" indicate movement of proteins from the Golgi to *"Undefined"* or the *"Plasma Membrane"*. **b** Schematic of plasma membrane proteomics workflow. Created in BioRender. Conway, O. (2026) BioRender.com/1rcx89m. **c** Scatter plot of changes in protein abundance at the 3T3-L1 adipocyte PM following insulin stimulation. *P*-values were calculated using Significance A values and corrected for multiple hypothesis testing using the Benjamini-Hochberg method. **d** PCA projection of LOPIT-DC adipocyte subcellular map under basal (left) and insulin-stimulated (right) conditions, highlighting C3ORF18 localisation (grey diamond). **e** Distribution of PM marker proteins (pink = basal, blue = insulin) and C3ORF18 in LOPIT-DC data under basal (solid black line) and insulin-stimulated (dashed black line) conditions. Mean profile of C3ORF18 shown, *n* = 3 biological replicates per condition. Source data are provided as a Source Data File.

0.99, 0.1, 0.4 respectively; Fig. 2e). Notably, DEPDC5, WDR59 and SZT2 exhibited a profile shift consistent with dissociation from lysosomes in response to insulin (Fig. 2e). Together, these data suggest that protein phosphorylation and redistribution work in concert to mediate adipocyte insulin signalling, and highlight the utility of assessing both post-translational modification and protein localisation in studying mechanisms of cell signalling.

Membrane trafficking terms were also over-represented in differentially localised proteins (Fig. 2c, Supplementary Data 1). These included proteins implicated in GLUT4[11,52,53] (e.g., RALA, STX16, STX6, CDC42, PICALM, VTI1B, SCAMP1) and endosomal traffic (e.g., SNX12, VPS26A, SNX27, STX12) (Supplementary Data 1). We therefore assessed the localisation of cytosolic protein complexes that regulate cargo sorting through this system (e.g., retromer, ESCRT, AP-1, AP-2, AP-3, COPI, COPII, HOPS). These complexes had distinct distributions, and subunits within each complex clustered tightly when visualised by PCA (Fig. 2f, Supplementary Fig. 2e). Specific complexes had particularly striking relocalisation in response to insulin, including retromer (VPS26A/B-VPS29-VPS35, *mean subunit diff. loc. prob.* = 0.31; Fig. 2g) and AP-3 (*mean subunit diff. loc. prob.* = 0.77, Fig. 2g). However, others such as ESCRT-II and AP-1 did not alter their subcellular distribution (*mean subunit diff. loc. prob.* = 0, Fig. 2g). Together, these data suggest insulin signalling targets discrete components of the endolysosomal system.

### The plasma membrane proteome undergoes dynamic remodelling in response to insulin

LOPIT-DC revealed the adipocyte PM proteome to be highly insulin responsive (Fig. 3a, Supplementary Fig. 3a). To provide a more

quantitative assessment of protein abundance changes at the PM in response to insulin, we used an orthologous cell surface biotinylation proteomics approach[13] (Fig. 3b). Briefly, 3T3-L1 adipocytes were left untreated or stimulated with insulin (100 nM, 20 min) before surface proteins were biotinylated and isolated by affinity purification. Protein identification and relative cell surface abundance were determined by LC-MS. 342 proteins exhibited altered abundance at the PM following insulin stimulation (119 increased; 223 decreased; Fig. 3c, Supplementary Data 2). These data highlight a substantial remodelling of the adipocyte PM proteome in response to acute insulin stimulation. We identified known insulin-responsive PM proteins GLUT1[54], GLUT4[55–57], LNPEP/IRAP[50,51], IGF2R/CI-M6PR[6,7] and TfR[4], which all increased in abundance at the PM, and INSR[58], which decreased (Fig. 3c). Twenty-seven proteins that changed in abundance at the PM in response to insulin (*p* < 0.05) were also predicted to differentially localise with high-confidence in our whole cell subcellular proteomics data (*diff. loc. prob.* = 1) (Supplementary Fig. 3b, Supplementary Data 2). Within these, C3ORF18, an uncharacterised protein of unknown function, had one of the greatest (1.8-fold) increases in abundance at the PM in response to insulin (Supplementary Data 2). Accordingly, in our LOPIT-DC data, C3ORF18 also relocalised in response to insulin (*diff. loc. prob.* = 1, Fig. 3d), with greater enrichment in fractions containing PM marker proteins after insulin stimulation (Fig. 3e).

### C3ORF18 redistributes to the plasma membrane in response to insulin

The type III membrane protein C3ORF18 captured our attention as it underwent insulin-stimulated redistribution in both our whole-cell spatial proteomics and PM proteomic profiling, and was upregulated

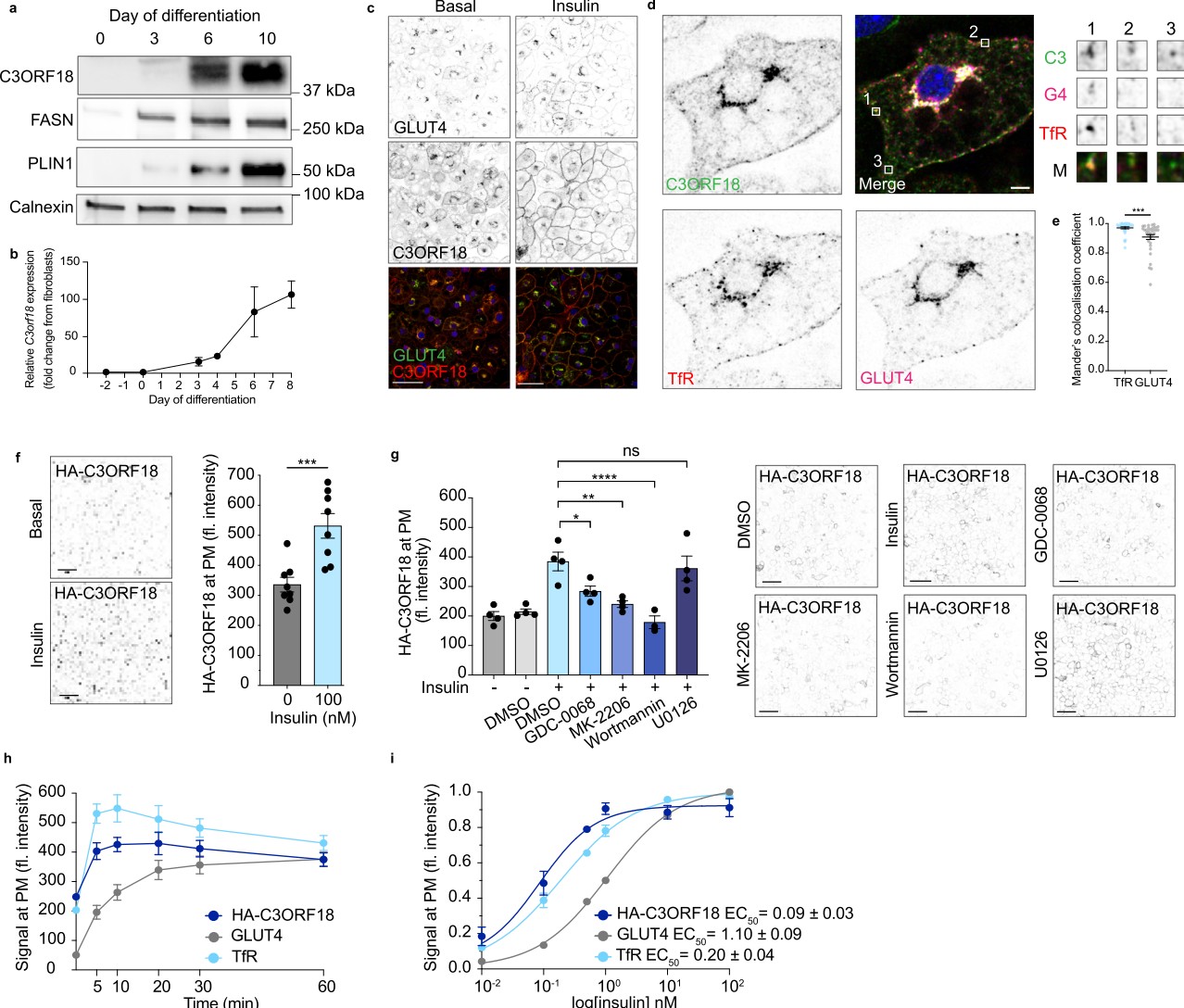

**Fig. 4 | C3ORF18 is an insulin-responsive protein. a** Western blot analysis of C3ORF18 protein abundance over differentiation in 3T3-L1 adipocytes. Representative blot, n = 2 biological replicates. **b** *C3orf18* expression over differentiation in 3T3-L1 adipocytes (expression normalised to *Ppia*), n = 3 biological replicates. **c** Immunostaining of C3ORF18 in permeabilised 3T3-L1 adipocytes (scale bar = 50 μm) under basal and insulin-stimulated (100 nM, 30 min) conditions. Data are representative of 3 independent experiments. **d** Representative image of C3ORF18, TfR and GLUT4 immunostaining in 3T3-L1 adipocytes (scale bar = 10 μm) and **e** quantification of colocalisation of C3ORF18 with TfR and GLUT4 in (**d**) using Manders colocalisation coefficient in ImageJ, each data point represents an individual cell (n = 35 cells, p = 0.0005). **f** Anti-HA surface immunostaining (left) and relative fluorescence intensity (right) in basal and insulin-stimulated (100 nM, 30 min) non-permeabilised 3T3-L1 adipocytes (n = 8 biological replicates, scale bar = 100 μm, p = 0.0003). **g** Relative fluorescence intensity of surface HA-C3ORF18

in 3T3-L1 adipocytes treated with AKT-PI3K (GDC-0068, MK-2206, Wortmannin) or MAPK (U0126) signalling inhibitors prior to insulin stimulation (100 nM, 30 min; n = 4 biological replicates for 0 nM insulin, DMSO, GDC-068, MK-2206, U0126; n = 3 for biological replicates for Wortmannin; DMSO vs. GDC-0068 p = 0.034, MK-2206 p = 0.002, Wortmannin p < 0.0001, U0126 p = 0.966). Representative images for each condition on the right. **h** Relative fluorescence intensity of surface HA-C3ORF18 (dark blue), GLUT4 (grey) and TfR (light blue) in response to 100 nM insulin for 0–60 min in adipocytes (n = 7 biological replicates). **i** Relative fluorescence intensity of surface HA-C3ORF18 (dark blue), GLUT4 (grey) and TfR (light blue) in adipocytes following 30 min stimulation with increasing doses of insulin (0–100 nM, n = 7 biological replicates). All data represented as mean ± SEM (**b**, **e–i**). n.s. non-significant; *p < 0.05; **p < 0.01; ***p < 0.01; ****p < 0.0001 by paired two-tailed Student's t-test (**e**, **f**) or one-way ANOVA with Šidák's multiple comparisons test (**g**). Source data are provided as a Source Data File.

at mRNA and protein levels during adipocyte differentiation (Fig. 4a, b). Despite a predicted molecular weight of ~17 kDa, we detected an anti-C3ORF18 immunoreactive band at ~40 kDa following SDS-PAGE and immunoblot (Fig. 4a). We validated this ~40 kDa band as C3ORF18 by 1) observing the same apparent molecular weight with overexpressed epitope-tagged C3ORF18 (Supplementary Fig. 4a); 2) showing that siRNA-mediated depletion of C3ORF18 led to loss of this ~40 kDa band (Supplementary Fig. 4b); and 3) demonstrating this ~40 kDa band was insulin responsive in the PM fraction (Supplementary Fig. 4c). Additional biochemical studies revealed that the

N-terminal luminal portion of C3ORF18 was responsible for its reduced electrophoretic mobility, perhaps because of high disorder in this region[59] (Supplementary Fig. 4d–g).

To assess C3ORF18 subcellular localisation and validate the insulin-responsive relocalisation observed in our proteomics (Fig. 3c–e) and Western blot data (Supplementary Fig. 4c), we immunostained endogenous C3ORF18 in adipocytes under basal and insulin-stimulated conditions. This revealed C3ORF18 localised to both the perinuclear region and cell periphery, with increased peripheral staining in response to insulin (Fig. 4c, d). We observed similar

localisation in adipocytes over-expressing HA-C3ORF18 (Supplementary Fig. 4h). HA-C3ORF18 did not colocalise with the ER-marker calnexin, and the perinuclear staining of HA-C3ORF18 was juxtaposed to the trans-Golgi marker TGN46 (Supplementary Fig. 4i), suggesting this is a post-Golgi compartment. HA-C3ORF18 had limited overlap with the early endosome marker EEA1 (Supplementary Fig. 4i), but endogenous C3ORF18 substantially colocalised with TfR and GLUT4, particularly in the perinuclear region (Fig. 4d). Quantitative analysis revealed greater colocalisation of C3ORF18 with TfR than GLUT4 (Fig. 4e), best exemplified when assessing C3ORF18 and TfR immunostaining in punctate structures towards the cell periphery under basal conditions (Fig. 4d). Importantly, this analysis also identified that C3ORF18 can be found in vesicles that are both TfR- and GLUT4-negative (Fig. 4d, inset). This suggests that C3ORF18 may reside, at least partially, in a distinct vesicle population from both TfR and GLUT4.

We next compared the C3ORF18 translocation responses to those of the known insulin-regulated proteins TfR and GLUT4. To more accurately quantify the insulin-responsive translocation of C3ORF18 to the PM, we used N-terminally HA-tagged C3ORF18 (HA-C3ORF18) expressing cell lines, where the HA-epitope is luminal/extracellular, to specifically label PM-localised C3ORF18. Surface staining in non-permeabilised cells confirmed insulin-responsive translocation of HA-C3ORF18 to the PM in 3T3-L1 adipocytes (Fig. 4f), SGBS adipocytes and L6 myotubes and myoblasts (Supplementary Fig. 4j). In addition, insulin-stimulation increased plasma membrane C3ORF18 in primary rat adipocytes (Supplementary Fig. 4k). As for GLUT4 and TfR, HA-C3ORF18 translocation was not unique to insulin, as 5 min EGF-stimulation also increased PM abundance (Supplementary Fig. 4l). Small molecule inhibition of PI3K (Wortmannin) and AKT (GDC-0068, MK-2206), but not MAPK (U0126), impaired insulin-stimulated HA-C3ORF18 PM translocation (Fig. 4g, Supplementary Fig. 4m), demonstrating C3ORF18 relocalisation is dependent on the PI3K-AKT branch of insulin signalling, as previously described for both GLUT4 and TfR[60] (Supplementary Fig. 4n-o). Consistent with our co-localisation data (Fig. 4d, e), the dose response and kinetics of C3ORF18 translocation to the PM were more closely aligned with TfR than those of GLUT4 (Fig. 4h, i). For example, the $t_{1/2}$ for C3ORF18 and TfR was <1 min and 1 min, respectively, but 6.2 min for GLUT4. Collectively, these data suggest C3ORF18 redistributes to the PM from an insulin/growth factor-responsive, likely TfR-positive, endosomal compartment.

### C3ORF18 is required for maximal insulin sensitivity

Having validated the insulin-responsive redistribution of C3ORF18 to the PM, we next determined whether this protein played a functional role in adipocyte biology. Although C3ORF18 was upregulated during differentiation (Fig. 4a, b), *C3orf18* depletion from preadipocytes and during differentiation did not affect 3T3-L1 differentiation as measured by lipid accumulation or expression of adipocyte marker genes 8 d after initiation of differentiation (Supplementary Fig. 5a–d).

To study whether C3ORF18 plays a role in the adipocyte insulin response, we depleted *C3orf18* in differentiated adipocytes (siRNA introduced 6 d after initiation of differentiation), with C3ORF18 protein abundance ~60% and ~90% of control cells after 4 (10 d after initiation of differentiation) and 8 d (14 d after initiation of differentiation) knockdown, respectively (Supplementary Fig. 5e). siRNA-mediated depletion of *C3orf18* for 8 d reduced insulin-stimulated 2-deoxyglucose transport in human and mouse adipocytes (Fig. 5a, b, Supplementary Fig. 5f, g), and, consistent with this, impaired insulin-stimulated GLUT4 translocation to the PM (Fig. 5c). Impaired insulin responses were not specific to glucose transport, as insulin-stimulated TfR translocation to the cell surface was also reduced (Fig. 5d). To determine clinical relevance of these observations, we assessed the correlation between human adipose tissue *C3ORF18* expression and clinical metabolic traits using the Adipose Tissue Knowledge Portal[26].

*C3ORF18* expression was lower in subjects with higher circulating blood glucose and HbA1c (Fig. 5e), and increased with weight loss (Supplementary Fig. 5h). These clinical correlative data are directionally consistent with our data from cultured cells, indicating that C3ORF18 positively regulates insulin sensitivity in adipocytes.

Decreased insulin-stimulated 2-deoxyglucose transport after 8 d *C3orf18* knockdown was likely driven by both lower protein abundance of the glucose transporters GLUT4 and GLUT1, independent of mRNA expression (Fig. 5f, Supplementary Fig. 5i), as well as reduced proximal insulin signalling (Fig. 5g, Supplementary Fig. 5j). Lower insulin signalling responses were independent of changes in INSR mRNA and protein abundance or localisation (Supplementary Fig. 5k–m), but were associated with decreased IRS1 protein, as well as reduced expression of adipocyte marker genes (Supplementary Fig. 5n–p).

To identify the earliest phenotypic responses to *C3orf18* depletion from differentiated adipocytes, we assessed insulin responses after 4 d knockdown, when C3ORF18 protein levels were ~60% of control cells (Supplementary Fig. 5e). At this timepoint, insulin-stimulated 2-deoxyglucose, TfR translocation and TfR protein abundance were reduced in *C3orf18* knockdown cells (Supplementary Fig. 5q–t). Insulin signalling and insulin-stimulated GLUT4 translocation were not affected (Supplementary Fig. 5u, v). These data suggest that endosomal recycling/protein stability is sensitive to C3ORF18 depletion, consistent with C3ORF18 subcellular localisation (Fig. 4d, e).

## Discussion

Here, we present a subcellular map of the 3T3-L1 adipocyte proteome under basal conditions and following acute insulin stimulation, to provide an unbiased cell-wide analysis of insulin-stimulated protein relocalisation in adipocytes. Combined with a quantitative analysis of the PM, these data revealed extensive protein redistribution in response to insulin. We highlight the utility of this dynamic subcellular mapping to identify regulators of adipocyte insulin action by studying C3ORF18, which traffics to the PM in response to insulin and is required for maintaining insulin sensitivity in adipocytes.

Previous work has focused on insulin-stimulated protein translocation events in individual organelles, such as GLUT4 storage vesicles[11,53] or the PM[8], but no studies to date have taken an unbiased whole-cell approach used herein. Approximately ~10% (502 proteins) of identified proteins exhibited high-confidence ($diff. loc. prob. = 1$) subcellular redistribution in response to insulin, with the PM proteome particularly dynamic. The significant remodelling of the PM proteome in response to insulin is, to some extent, unsurprising, since insulin-regulated delivery of GLUT4 (and proteins that colocalise with GLUT4) to the PM is highly studied[3]. However, a large proportion of proteins that increased in abundance at the PM with insulin are not reported to localise to specialised GLUT4 storage vesicles[11,52,53], suggesting that other vesicular carriers are insulin responsive. Indeed, endosomal cargoes, including IGF2R/CI-M6PR[6,7], TfR[4,5], also undergo insulin-responsive trafficking to the PM. Further, we report here that C3ORF18 is a insulin-responsive protein, with the high degree of colocalisation with TfR, and well as similar translocation kinetics in response to insulin, suggesting that C3ORF18 is delivered to the plasma membrane from a TfR-positive intracellular compartment, likely endosomes (Fig.4, Supplementary Fig. 4). The extent of the endosomal response to insulin, why this occurs and how this is regulated remains unclear. Our data suggest that there is redistribution of several endosomal cargoes to the PM in response to insulin, and that insulin signalling specifically regulates a subset of endomembrane trafficking complexes which directly bind cargoes (for example retromer and AP-3, Fig. 2, Supplementary Fig. 2). These data are in line with the growing recognition of the role the endosomal network plays in metabolic regulation and pathologies[61], and provide a foundation for studies into the spatial regulation of endolysosomal trafficking by insulin/growth factor signalling.

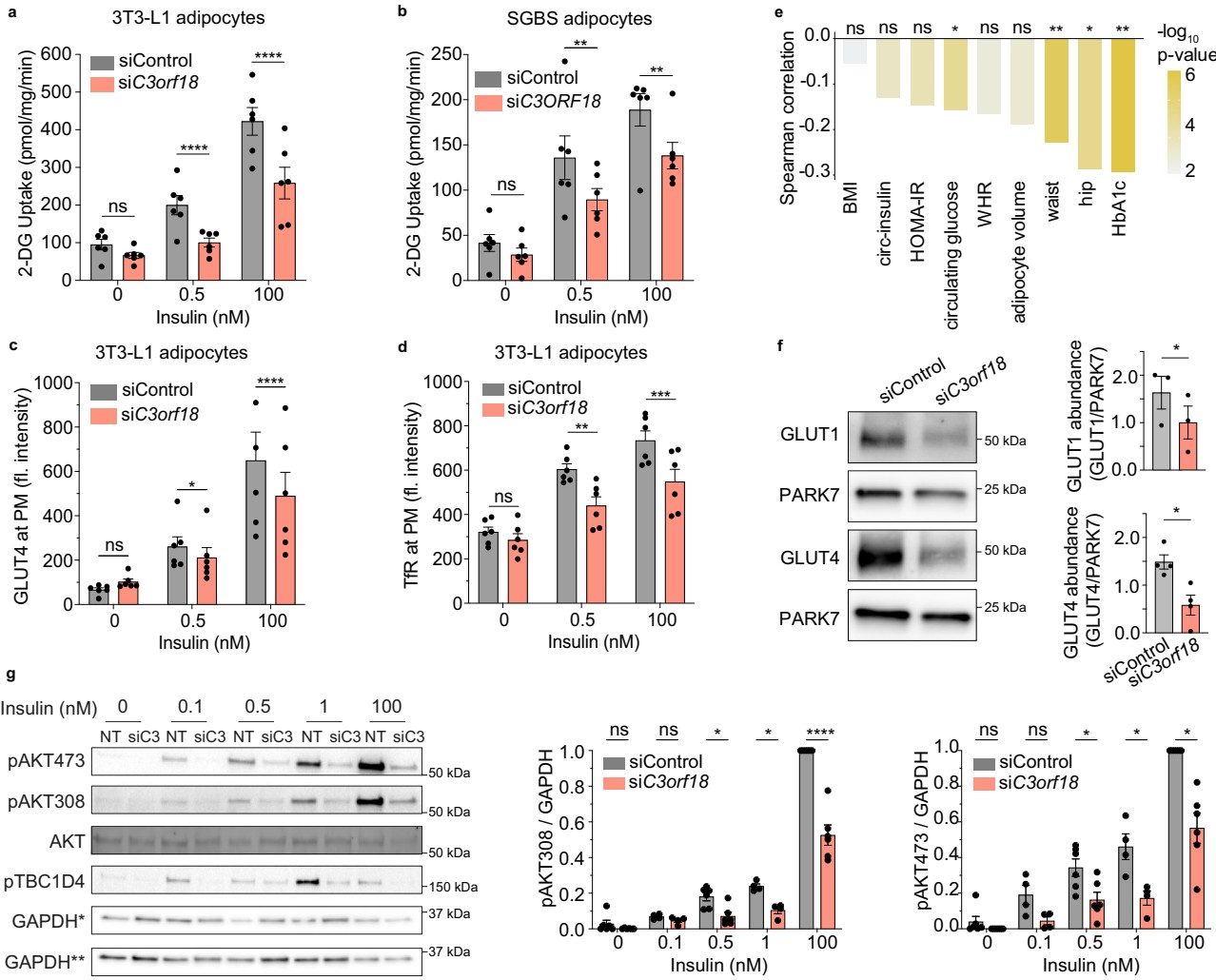

**Fig. 5 | C3ORF18 is required for maximal insulin sensitivity. a, b** 2-deoxyglucose (2-DG) uptake in 3T3-L1 (**a**) and SGBS (**b**) adipocytes following 8 d *C3ORF18* depletion (pale red, non-targeting control siRNA in grey; $n = 6$ biological replicates; 3T3-L1 0 nM $p = 0.134$, 0.5 nM $p < 0.0001$, 100 nM $p < 0.0001$; SGBS 0 nM $p = 0.627$, 0.5 nM $p = 0.007$, 100 nM $p = 0.004$). **c, d** Relative fluorescence intensity of surface GLUT4 (**c**) and TfR (**d**) in 3T3-L1 adipocytes treated with non-targeting (siControl) or *C3orf18* siRNA for 8 d and stimulated with 0.5 or 100 nM insulin for 30 min ($n = 6$ biological replicates; GLUT4 0 nM $p = 0.165$, 0.5 nM $p = 0.041$, 100 nM $p < 0.0001$; TfR 0 nM $p = 0.699$, 0.5 nM $p = 0.001$, 100 nM $p = 0.0003$). **e** Two-sided Spearman's correlation of *C3ORF18* mRNA expression in omental adipose with metabolic clinical features. Figure generated from adiposetissue.org[26] using data from[27–47] (** pFDR < 0.01, * pFDR < 0.05). **f** Representative Western blot and quantification of GLUT1 and GLUT4 protein expression in 3T3-L1 adipocytes following 8 d *C3orf18* knockdown ($n = 4$ biological replicates; GLUT1 $p = 0.049$, GLUT4 $p = 0.014$).

**g** Representative Western blot of phosphorylated AKT (Thr308 and Ser473) and TBC1D4 (Thr642) in 3T3-L1 adipocytes following 8 d *C3orf18* depletion, stimulated with 0, 0.1, 0.5, 1 or 100 nM insulin for 30 min. GAPDH* loading control for total AKT and pAKT308, GAPDH** loading control for pAKT473 and pTBC1D4, NT is non-targeting siRNA control. Densitometric quantification of phosphoAKT normalised to GAPDH loading controls in right panels ($n = 6$ biological replicates for 0, 0.5 and 100 nM; $n = 4$ for 0.1 and 1 nM insulin; pAKT308 0 nM $p = 0.954$, 0.1 nM $p = 0.9998$, 0.5 nM $p = 0.027$, 1 nM $p = 0.037$, 100 nM $p < 0.0001$; pAKT473 0 nM $p = 0.754$, 0.1 nM $p = 0.341$, 0.5 nM $p = 0.026$, 1 nM $p = 0.042$, 100 nM $p = 0.018$). All data represented as mean ± SEM unless otherwise stated. n.s. non-significant; *$p < 0.05$; **$p < 0.01$; ***$p < 0.001$; ****$p < 0.0001$ by paired two-tailed Student's $t$-test (**f**) or two-way ANOVA with Sidak's multiple comparisons test (**a**–**d**, **g**). Source data are provided as a Source Data File.

C3ORF18 was not required for adipocyte differentiation. However, prolonged (8 d) depletion of *C3orf18* from adipocytes impaired insulin-stimulated 2-deoxyglucose transport and delivery of both TfR and GLUT4 to the PM. This appears to be driven by both lower protein abundance of TfR and the glucose transporters GLUT4 and GLUT1, and impaired insulin signalling to AKT (Fig. 5, Supplementary Fig. 5). Directionally consistent with these data, *C3ORF18* expression in omental adipose tissue was negatively correlated with circulating glucose and HbA1c. Further, the link between C3ORF18 and adipocyte biology is supported by human genetics, where a common variant, rs1034405, is associated with increased waist-to-hip ratio adjusted for BMI[62]. The molecular mechanisms linking C3ORF18 to adipocyte insulin responses remain unclear, although our data from cells

depleted of *C3orf18* for shorter periods suggest the primary effect of loss of C3ORF18 is an impairment in endosomal trafficking, as indicated by loss of TfR protein and impaired TfR delivery to the PM. Our working model is that C3ORF18 promotes proper endosomal recycling, and impairing this process over longer periods disrupts adipocyte insulin responses via a range of mechanisms, including 1) direct effects on membrane traffic; 2) loss of key insulin-responsive proteins that traffic via the endosomal system (i.e., GLUT4); and 3) transcriptional responses that suggest a role for C3ORF18 in maintaining mature adipocyte identity. Future studies will need to address the exact mechanisms underpinning insulin-stimulated C3ORF18 redistribution and C3ORF18 linking C3ORF18 and endosomal traffic to longer-term adipocyte insulin sensitivity, and the role C3ORF18 plays in other

tissues where it is enriched, such as muscle. Nonetheless, this example demonstrates that changes in protein subcellular location can be used to identify functionally relevant proteins.

The scale of protein redistribution we identified highlights the likely importance of protein localisation in the adipocyte insulin responses. Proteins predicted to relocalise in response to insulin were often also subject to insulin-regulated phosphorylation, suggesting that, in some cases, these two mechanisms of protein regulation may be linked. As an example, mTOR itself, as well as subunits of the KICSTOR and GATOR complexes implicated in the regulation of mTORC1 activity at the lysosomes, were both phosphorylated and moved away from lysosomes in response to insulin. As these complexes are primarily implicated in amino acid sensing, this raises the possibility that insulin signalling directly regulates mTORC1 activity via phosphorylation-driven redistribution of GATOR/KICSTOR, as recently reported for GATOR2 by AMPK signalling[63]. In this way, the combination of dynamic protein localisation and phosphorylation data, and structural information if available (e.g., for the KICSTOR-GATOR1-SAMTOR complex[64]), may reveal mechanisms governing the cellular insulin response.

There are several limitations to our study. Although the 3T3-L1 adipocyte model has been extensively used to study adipocyte cell biology, these cells are derived from mice and so may not fully recapitulate protein localisation in human adipocytes. However, our comparison with localisations reported in human SGBS adipocytes indicated a high degree of similarity in protein localisation between mouse and human adipocytes. Further, we validated key observations on septins 2 and 11 and C3ORF18 in human cells. We employed strict thresholding of allocations to avoid over-allocation of multi-localised proteins to single organelle(s), and endosomes and the lipid droplet were removed as organelles for allocation, likely contributing to the lower number of proteins allocated to a single organelle in our study compared to others[20]. Importantly, detection of insulin-stimulated protein relocalisation was independent of protein allocation to an organelle, a key benefit of BANDLE. Indeed, a majority of insulin-responsive proteins were not allocated to a specific organelle and were of 'undefined' localisation under both conditions. These are likely multi-localised proteins, where a change in protein distribution with insulin represents changes in the relative abundance of the protein between subcellular sites. Finally, we generated data from two distinct subcellular proteomics approaches. Since these data were not entirely overlapping, there are likely many additional PM translocating proteins identified in our PM proteomic analysis that are worthy of follow-up, but were not considered here if not present in both analyses.

In summary, we present a high-resolution subcellular map of the adipocyte proteome and demonstrate the extent and importance of insulin-stimulated protein relocalisation in the adipocyte insulin response. These data are available through an interactive online resource (https://proteome.shinyapps.io/adipocyte2025/) for further mining by the research community. Finally, we demonstrate that C3ORF18 is required to maintain insulin sensitivity in adipocytes, illustrating the utility of studying acute protein relocalisation to identify regulators of insulin action.

## Methods

### Ethics statement
For studies using animals, ethical approval was granted by the University of Bath Animal Welfare and Ethical Review Body, and all procedures were conducted in accordance with UK Home Office regulations. All procedures were conducted in accordance with relevant guidelines and regulations.

### Cell culture
**3T3-L1 fibroblast culture and differentiation into adipocytes.** 3T3-L1 fibroblasts obtained from David James (University of Sydney),

originally from Howard Green (Harvard Medical School, Boston, MA), were maintained in Dulbecco's modified Eagle's medium (DMEM; Sigma) supplemented with 10% FBS (Gibco) and 2 mM GlutaMAX (Gibco) at 37 °C with 10% $CO_2$. Cells were passaged at ~60% confluence. Differentiation was induced at 100% confluence by the addition of 350 nM insulin, 0.5 mM 3-isobutyl-1-methylxanthine (IBMX), 250 nM dexamethasone and 400 nM biotin to the supplemented DMEM for 3 d, followed by 350 nM insulin for a further 3 d. Adipocytes were used 10–11 d following the initiation of differentiation.

**SGBS preadipocyte culture and differentiation into adipocytes.** SGBS adipocytes were provided by Professor Martin Wabitsch (Division of Paediatric Endocrinology and Diabetes, University Medical Centre Ulm, Germany), and were cultured and differentiated according to the established protocol[65,66]. Briefly, SGBS fibroblasts were maintained in DMEM/F-12 (Sigma) supplemented with 10% FBS, 2 mM GlutaMAX, 33 μM biotin and 17 μM pantothenate. Differentiation was induced at 100% confluence by the addition of 0.01 mg/mL transferrin, 20 nM insulin, 100 nM cortisol, 0.2 nM T3, 25 nM dexamethasone, 250 μM IBMX, and 2 μM rosiglitazone to serum-free DMEM/F-12 for 4 d. Adipocytes were maintained in post-differentiation media containing 0.01 mg/mL transferrin, 20 nM insulin, 100 nM cortisol and 0.2 nM T3 in serum-free DMEM/F-12 until use at day 12–14 after initiation of differentiation.

**L6 myoblast culture and differentiation into myotubes.** L6 myoblasts obtained from David James (University of Sydney) were maintained in AlphaMEM (Gibco) supplemented with 10% FBS and 2 mM GlutaMAX. Differentiation was induced at 80–90% confluence by culture in AlphaMEM supplemented with 2% horse serum (PAN-Biotech, P30-0702) and 2 mM GlutaMAX, and media were changed every 2 days. Cells were used 5–7 days following the onset of differentiation.

**Generation of HA-C3ORF18 overexpressing stable cell lines.** Empty vector or HA-C3ORF18-overexpressing 3T3-L1 and L6 cell lines were generated via retroviral transduction (pMMLV; VectorBuilder), as previously described[67]. Retrovirus was generated using the Plat-E packaging cell line[68]. HA-C3ORF18-overexpressing SGBS cells were generated via lentiviral transduction (pLV; VectorBuilder). Lentivirus was generated using the LentiX 293 T packaging cell line (Takara Bio, 632180). Transduced cells were selected with 2 μg/mL puromycin and maintained and differentiated as described above. Puromycin was removed prior to cell differentiation. Plasmids are described in Supplementary Data 3.

**HEK cell culture and transient HA-C3ORF18 overexpression.** Wild-type HEK-293 cells were cultured in high-glucose DMEM supplemented with 10% FBS (Gibco) and 2 mM GlutaMAX (Gibco) at 37 °C with 5% $CO_2$. Cells were seeded at $3 \times 10^5$ cells/well into a 24-well plate 24 h prior to transfection with 0.5 μg DNA delivered using Lipofectamine2000 (Invitrogen). Cells were harvested for immunoblotting 24 h following transfection as described below. Plasmids are described in Supplementary Data 3.

### LOPIT-DC
**Subcellular fractionation.** Differentiated 3T3-L1 adipocytes were cultured in low media volume as previously described for 48 h prior to the experiment (8 mL/10 cm dish, media changed every 24 h) to improve insulin sensitivity[69]. serum starved in basal DMEM (high glucose DMEM, GlutaMAX, 0.2% BSA) for 2 h and maintained in basal medium or stimulated with 100 nM insulin for 30 min. Four confluent 10 cm dishes were used per replicate per condition. Cells were washed twice with ice-cold PBS and harvested in homogenisation buffer (250 mM sucrose, 10 mM 4-(2-hydroxyethyl)-1-piperazineethanesulfonic acid (HEPES), pH 7.4, 2 mM EDTA, 2 mM magnesium acetate tetrahydrate)

containing EDTA-free phosphatase and protease inhibitor cocktail (Pierce). Cells were homogenised by 10 passes through a 21-gauge needle followed by 10 passes through a 27-gauge needle prior to centrifugation at 100 x g for 5 min at 4 °C to remove unlysed cells. Cells were fractionated by centrifugation in a benchtop Beckman Allegra X-15R centrifuge (fraction 1-3), a benchtop Eppendorf 5415 R centrifuge (fraction 4–5), and a Beckman Optima LE-80K ultracentrifuge with a TLA-55 rotor (fraction 6–8) at 4 °C (Supplementary Data 1) and pellets were kept on ice.

Pellets were resuspended in solubilisation buffer (8 M urea, 50 nM HEPES pH 7.4, 0.2% SDS) and sonicated using a tip probe sonicator. Supernatant from the last centrifugation step was retained as the cytosolic fraction, precipitated with ten volumes of cold acetone overnight at −20 °C, and centrifuged at 16,000 x g for 20 min at 4 °C. Supernatant was removed and the pellet air dried for 15 min before resuspension in solubilisation buffer.

**Lipid droplet isolation.** Lipid droplets were isolated using a modified version of the protocol described by Brasaemle and Wolins[70]. 3T3-L1 adipocytes were serum-starved in basal DMEM for 2 h, and treated with 100 nM insulin for 30 min as described above. Six confluent 10 cm plates of adipocytes were used per condition. Cells were washed twice with ice-cold PBS and harvested in Tris-EDTA buffer (20 mM Tris-HCl (pH 7.4), 1 mM EDTA, Complete protease inhibitor (Merck)), and disrupted by nitrogen cavitation (Parr Instruments). Cells were exposed to 41 Barr of nitrogen for 20 min, after which the pressure was slowly released and the cell homogenate collected dropwise. Homogenate was spun at 1000 x g for 10 min at 4 °C to remove unlysed cells, the supernatant was made up in 20% sucrose (w/v) in Tris-EDTA buffer and overlaid with 5% and 0% sucrose in Tris-EDTA buffer in a 4 mL ultra-centrifuge tube (Beckman Coulter). Samples were centrifuged at 28,000 x g for 30 min at 4 °C with no brake in a Beckman Coulter Optima LE-80K ultracentrifuge using a SW 40 Ti rotor (Beckman Coulter). Lipid droplets were harvested from the top of the gradient, washed once and resuspended in Tris-EDTA buffer. Lipid droplet-associated proteins were precipitated using chloroform-methanol, and the protein pellet was air-dried, before being resuspended in 100 μL solubilisation buffer and incorporated into the LOPIT-DC workflow.

**Proteomic sample preparation.** 100 μg of protein from each fraction was reduced with a final concentration of 10 mM DTT at 37 °C for 1 h, then alkylated with a final concentration 40 mM chloroacetamide (CAA) for 2 h at RT. Protein was precipitated with acetone overnight at −20 °C, and pellets were resuspended in HEPES, pH 8.5. Samples were digested with sequencing-grade trypsin (Promega) with a final enzyme:protein ratio of 1:40 for 1 h at 37 °C, followed by a second trypsinisation step at 1:20 overnight at 37 °C. Peptides were quantified using a fluorometric peptide assay (Thermo Fisher Scientific) according to the manufacturer's instructions.

50 μg of each fraction was labelled using a 10-plex tandem mass tag (TMT) kit (Thermo Fisher Scientific, Supplementary Data 1). TMT tags were equilibrated to RT and resuspended in 82 μL of LC-MS grade acetonitrile (ACN). 100 μL of each fraction (at concentration 0.5 μg/μL) was added to 41 μL of the appropriate tag. Labelling was allowed to proceed for 2 h on a shaker at RT. Reactions were quenched by adding 5% hydroxylamine and incubating for a further 45 min on a shaker at RT. The 10 fractions of each sample were then pooled into 1 LoBind Eppendorf and lyophilised using a SpeedyVac.

Samples were desalted using a C18 SepPak cartridge (100 mg sorbent, Waters). Cartridges were equilibrated with 2 washes of 100% ACN, 1 wash of 0.05% acetic acid and 3 washes of 0.1% trifluoroacetic acid (TFA). Samples were resuspended in TFA, and the pH was adjusted to <3 before loading on column, and washed 3 times with 0.1% TFA, twice with 0.05% acetic acid, eluted in 70% ACN and 0.05% acetic acid and lyophilised.

Samples were pre-fractionated offline using reverse-phase UPLC in an Acquity UPLC System with a diode array detector (Waters) using an Acquity UPLC bridged ethyl hybrid C18 column (2.1-mm ID × 150-mm; 1.7-μm particle size; Waters). Peptide fractions were collected over a 50 min linear gradient, combined into 15 concatenated fractions, and lyophilised.

**LC-MS3.** MS analysis was performed using a Lumos Orbitrap mass spectrometer coupled to a Dionex Ultimate 3000 RSLC nanoUPLC system (Thermo Fisher Scientific, Waltham, MA, USA). Peptides were loaded onto a pre-column (Thermo Fisher Scientific PepMap 100 C18, 5 mm particle size, 100° A pore size, 300 mm i.d. x 5 mm length) from the Ultimate 3000 auto-sampler with 0.1% formic acid (FA) for 3 min at a flow rate of 15 μL/min. After this period, the column valve was switched to allow elution of peptides from the pre-column onto the analytical column. Separation of peptides was performed by C18 reverse-phase chromatography at a flow rate of 300 nL/min using a Thermo Fisher Scientific reverse-phase nano Easy-spray column (Thermo Fisher Scientific PepMap C18, 2 mm particle size, 100° A pore size, 75 μm i.d. x 50 cm length). Solvent A was water + 0.1% FA, and solvent B was 80% ACN, 20% water + 0.1% FA. The linear gradient employed was 2–40% B in 93 min (total LC run time was 120 min, including a high organic wash step and column re-equilibration).

The eluted peptides from the C18 column LC eluant were sprayed into the mass spectrometer by means of an Easy-Spray source (Thermo Fisher Scientific). All m/z values of eluting peptide ions were measured in an Orbitrap mass analyser, set at a resolution of 120,000 and were scanned between m/z 380–1500 Da. Data-dependent MS/MS scans (Top Speed) were employed to automatically isolate and fragment precursor ions by collision-induced dissociation (CID; normalised collision energy (NCE): 35%), which were analysed in the linear ion trap. Singly charged ions and ions with unassigned charge states were excluded from being selected for MS/MS, and a dynamic exclusion window of 70 s was employed. The top 10 most abundant fragment ions from each MS/MS event were then selected for a further stage of fragmentation by synchronous precursor selection (SPS) MS3 in the HCD high-energy collision cell using HCD (NCE: 65%). The m/z values and relative abundances of each reporter ion and all fragments (mass range from 100–500 Da) in each MS3 step were measured in the Orbitrap analyser, which was set at a resolution of 50,000. This was performed in cycles of 10 MS3 events before the Lumos instrument reverted to scanning the m/z ratios of the intact peptide ions, and the cycle continued.

**MS spectra processing and peptide and protein identification.** Raw LC-MS data files were processed with Proteome Discoverer v2.5 (Thermo Fisher Scientific) using the Mascot server 2.8.3 (Matrix Science). The MS data were run against a Swiss-Prot Mus musculus database (downloaded September 2019) and the common repository of adventitious proteins (cRAP, v1.0). Precursor and fragment mass tolerances were set to 10 ppm and 0.2 Da, respectively. Trypsin was set as the enzyme of choice, and a maximum of 2 missed cleavages was allowed. Fixed modifications were set to carbamidomethyl(C) TMT6plex (N-term) and TMT6plex (K), and variable modifications were set to carbamyl(N-term), carbamyl(K), carbamyl(R), deamidation(N,Q), oxidation (M) and TMT6plex (S/T). Percolator version 3.05 was used to assess the false discovery rate (FDR), and only high-confidence peptides were retained. Quantification of the MS3 reporter ions was performed within the Proteome Discoverer workflow using the Most Confident Centroid method for peak integration and integration tolerance of 20 ppm. Reporter ion intensities were adjusted to the batch-specific TMT reporter ion isotope distributions prior to processing (Supplementary Data 1). The mass spectrometry proteomics data have been deposited to the ProteomeXchange

Consortium via the PRIDE partner repository with the dataset identifier PXD061017.

**Spatial proteomics analysis.** All proteomics data analysis was performed in R (v4.3.2) using the Bioconductor packages pRoloc (v1.42.0)[19], bandle (v1.6)[18,48], clusterProfiler (v4.10.1)[71,72] and ggplot2 (v3.5.1).

Peptide spectrum match (PSM)-level quantitation output from Proteome Discoverer was imported into R, and non-specific filtering was performed as previously described[12,73] to retain high-quality PSMs. PSMs were sum normalised, and aggregated to protein level intensities using the 'robust' method[74] within the pRoloc R package[75]. Replicates were concatenated to yield 4005 proteins quantified across all 6 replicates, and allowed the generation of a single subcellular map for each condition. The data were annotated with marker proteins, i.e., proteins well established to localise to a specific subcellular compartment, from previous subcellular proteomics experiments[12,15,76]. Additional marker proteins were identified through examining protein localisation in UniProt for proteins reported only to be localised to one organelle, and by mining the literature. Proteins were manually excluded as marker proteins if they appeared to relocalise in response to insulin. Under de Duve's principle[77], proteins residing in the same subcellular compartment are expected to share characteristic fractionation profiles. BANDLE leverages this by comparing the profiles of proteins with unknown localisation to those of curated organelle markers to infer subcellular assignments.

To assess the resolution and the quality of separation between subcellular niches in the data, a QSep score was computed using the pRoloc package in R[17]. QSep calculates the Euclidean distance between raw protein profiles for each set of organelle markers, with higher values indicating better separation in a multi-dimensional space. Dimensionality reduction using principal component analysis was also performed to visually assess subcellular separation indicated by the presence of distinct clusters. The final marker list incorporated 531 proteins from 9 subcellular niches; cytosol, ER, Golgi apparatus, lysosomes, mitochondrion, nucleus, peroxisomes, PM, and ribosomes (Supplementary Data 1).

The Bayesian Analysis of Differential Localisation Experiments (BANDLE) algorithm from the bandle package[48] was used to assign proteins to subcellular niches and predict differential localisation with the following parameters; 30,000 MCMC iterations, 5000 burn-in iterations, 4 chains, and 1/20 thinning. Each protein was assigned to a subcellular niche for which it had the highest mean allocation probability, as described in the bandle vignette on Bioconductor (https://bioconductor.org/packages/release/bioc/html/bandle.html)[78]. Allocations were thresholded to retain only high-confidence allocations. Proteins were assigned to a subcellular niche if their allocation probability was >0.9 and their outlier probability was < the median outlier probability for that organelle. Proteins were deemed differentially localised if the differential localisation probability > 0. Proteins with differential localisation probability = 1 were considered 'high confidence' and all others as 'candidates'.

**Enrichment analysis.** Gene ontology[79] cellular compartment enrichment was performed in turn for proteins assigned to each organelle (not including marker proteins) using clusterProfiler[71]. Gene set enrichment analysis was performed using Gene Ontology biological process and KEGG[80] enrichment in clusterProfiler, and REACTOME[81] enrichment analysis was performed using reactomePA (v1.46). In all analyses, *p*-values were adjusted using the Benjamini-Hochberg procedure[82].

**Phosphoproteomics enrichment analysis.** Proteins in Humphrey (2013)[1] with phosphorylation sites with -/+ 2 fold change in response to insulin at any time point (0.25, 0.5, 1, 2, 5, 10, 20, 60 min) or proteins

from Fazakerley (2023)[10] with phosphorylation sites significantly up- or down-regulated in response to insulin were considered "insulin-regulated". This list was converted to mouse UniProt IDs using UniProt align, with only reviewed entries taken. Enrichment of insulin-stimulated phosphorylation in the proteins predicted to differentially localise vs. not was determined using a Fisher's exact test in R. Insulin-stimulated phosphorylation status of all proteins identified in LOPIT-DC in both phosphorylation data sets are shown in Supplementary Data 1.

**Plasma membrane proteomics**
Plasma membrane proteomics was performed as previously described[13] with minor modifications. Differentiated 3T3-L1 adipocytes were cultured in low media volume as described above for 48 h prior to the experiment, serum starved in basal DMEM for 2 h and maintained in basal medium or stimulated with 100 nM insulin for 30 min. Three confluent 10 cm dishes were used per replicate per condition, and six biological replicates were performed. Cells were washed twice with ice-cold PBS containing $MgCl_2$ and $CaCl_2$ (Sigma), and all subsequent steps were performed on ice. An oxidation/biotinylation mix comprising 200 mM aminooxybiotin (Biotium), 3 mM sodium meta-periodate (Thermo Fisher Scientific) and 10 mM aniline (Sigma) in ice-cold PBS pH 6.7 was added, and dishes were rocked on ice for 1 h at 4 °C in the dark. The reaction was quenched by adding glycerol to a final concentration of 1 mM for 5 min. Cells were washed twice with PBS pH 7.4 and lysed in a lysis buffer comprising 1.6% Triton X-100, 10 mM Tris-HCl pH 7.6, 150 mM NaCl, 5 mM iodoacetamide (IAA) and Complete protease inhibitor (Merck) for 30 min on ice. Nuclei (pellet) and lipid (top phase) were removed by centrifugation at 4 °C, 13,000 x *g* for 10 min, which was repeated three times. 5 mg of protein was incubated with high-affinity streptavidin agarose beads (Pierce, 20357, 10 μL beads/mg protein) at 4 °C for 18 h on a rotor to capture biotinylated glycoproteins. Beads were washed extensively with lysis buffer and PBS/0.5% SDS, using Poly-Prep columns (BioRad) attached to a vacuum manifold. Captured proteins were reduced with PBS/0.5% SDS/ 100 mM DTT for 20 min at RT, washed extensively with UC buffer (6 M urea in 0.1 M Tris-HCl pH 7.6), alkylated with 50 mM IAA (Sigma) for 20 min at RT, and digested on-bead with trypsin (Promega) in 200 mM HEPES pH 8.5 for 3 h at 37 °C. Tryptic peptides were collected and stored at -80 °C until TMT labelling.

**Peptide labelling with tandem mass tags.** TMT reagents (TMTpro 16-plex, 0.8 mg, Thermo Fisher Scientific) were dissolved in 43 μL anhydrous ACN and 10 μL added to the peptide at a final ACN concentration of 30% (v/v). Samples were labelled as in Supplementary Data 2, incubated at RT for 1 h, and reaction quenched with hydroxylamine to a final concentration of 0.5% (v/v). TMT-labelled samples were combined at a 1:1:1:1:1:1:1:1:1:1:1:1:1 ratio. The sample was vacuum-centrifuged to near dryness and subjected to C18-based solid phase extraction (Sep-Pak, Waters). An unfractionated 'single shot' was analysed initially to ensure similar peptide loading across each TMT channel, thus avoiding the need for excessive electronic normalisation. As all normalisation factors were >0.5 and <2, data for each single-shot experiment were analysed with data for the corresponding fractions to increase the overall number of peptides quantified. Normalisation is discussed in 'data analysis,' and high pH reversed-phase (HpRP) fractionation is discussed below.

**Offline HpRP fractionation.** TMT-labelled tryptic peptides were subjected to HpRP fractionation using an Ultimate 3000 RSLC UHPLC system (Thermo Fisher Scientific) equipped with a 2.1 mm internal diameter (ID) x 25 cm long, 1.7 μLm particle Kinetix Evo C18 column (Phenomenex). Mobile phase consisted of A: 3% ACN, B: 100 % ACN and C: 200 mM ammonium formate, pH 10. Isocratic conditions were 90% A/10% C, and C was maintained at 10% throughout the gradient elution.

Separations were conducted at 45 °C. Samples were loaded at 200 μL/min for 5 min. The flow rate was then increased to 400 μL/min over 5 min, after which the gradient elution proceeded as follows: 0-19% B over 10 min, 19–34% B over 14.25 min, 34–50% B over 8.75 min, followed by a 10 min wash at 90% B. UV absorbance was monitored at 280 nm and 15 s fractions were collected into 96 well microplates using the integrated fraction collector. Adjacent columns of fractions were combined, resulting in six combined fractions. Wells were excluded prior to the start or after the cessation of elution of peptide-rich fractions, as identified from the UV trace. Fractions were dried in a vacuum centrifuge and resuspended in 10 μL MS solvent (4% ACN/5% FA) prior to LC-MS3.

**LC-MS3.** Labelled samples were analysed using an Orbitrap Fusion Lumos (Thermo Fisher Scientific). An Ultimate 3000 RSLC UHPLC machine equipped with a 300 μm internal diameter × 5 mm Acclaim PepMap μ-Precolumn (Thermo Fisher Scientific) and a 75 μm internal dimeter × 50 cm 2.1 μm particle Acclaim PepMap RSLC analytical column was used. The loading solvent was 0.1% FA. The analytical solvent consisted of 0.1% FA (A) and 80% ACN + 0.1% FA (B). All separations were carried out at 40 °C. Samples were loaded at 5 μL/min for 5 min in loading solvent. For the single-shot sample, the analytical gradient consisted of 3–7% B over 3 min, 7–37% B over 173 min, followed by a 4 min wash at 95% B and equilibration at 3% B for 15 min. For fractionated samples, the analytical gradient consisted of 3–7% B over 4 min, 7–37% B over 116 min, followed by a 4 min wash at 95% B and equilibration at 3% B for 15 min. Each analysis used a MultiNotch MS3-based TMT method[83,84]. The following settings were used: MS1: 380-1500 Th, 120,000 resolution, $2 \times 10^5$ automatic gain control (AGC) target, 50 ms maximum injection time. MS2: quadrupole isolation at an isolation width of mass-to-charge ratio (m/z) 0.7, collision-induced dissociation fragmentation (normalised collision energy (NCE) 34) with ion trap scanning in turbo mode from m/z 120, $1.5 \times 10^4$ AGC target, 120 ms maximum injection time. MS3: in Synchronous Precursor Selection mode, the top 10 MS2 ions were selected for HCD fragmentation (NCE 45) and scanned in the Orbitrap at 60,000 resolution with an AGC target of $1 \times 10^5$ and a maximum accumulation time of 150 ms. Ions were not accumulated for all parallelisable time. The entire MS/MS/MS cycle had a target time of 3 s. Dynamic exclusion was set to ±10 ppm for 70 s. MS2 fragmentation was triggered on precursors $5 \times 10^3$ counts and above. The mass spectrometry proteomics data have been deposited to the ProteomeXchange Consortium via the PRIDE partner repository with the dataset identifier PXD061616.

**Data analysis.** Mass spectra were processed using a Sequest-based software pipeline for quantitative proteomics, 'MassPike', through a collaborative arrangement with Professor Steven Gygi's laboratory at Harvard Medical School. MS spectra were converted to mzXML using an extractor built upon Thermo Fisher's RAW File Reader library (version 4.0.26). In this extractor, the standard mzxml format has been augmented with additional custom fields that are specific to ion trap and Orbitrap mass spectrometry and essential for TMT quantitation. These additional fields include ion injection times for each scan, Fourier Transform-derived baseline and noise values calculated for every Orbitrap scan, isolation widths for each scan type, scan event numbers, and elapsed scan times. This software is a component of the MassPike software platform and is licensed by Harvard Medical School.

A combined database was constructed from the mouse Uniprot database (2019) and common contaminants such as porcine trypsin and endoproteinase LysC. The combined database was concatenated with a reverse database composed of all protein sequences in reverse order. Searches were performed using a 20-ppm precursor ion tolerance. Fragment ion tolerance was set to 1.0 Th. TMT tags on lysine residues and peptide N termini (304.2071 Da) and carbamidomethylation of cysteine residues (57.02146 Da) were set as static modifications, while oxidation of methionine residues (15.99492 Da) was set as a variable modification.

To control the fraction of erroneous protein identifications, a target-decoy strategy was employed[85]. Peptide spectral matches (PSMs) were filtered to an initial peptide-level FDR of 1% with subsequent filtering to attain a final protein-level FDR of 1%. PSM filtering was performed using a linear discriminant analysis, as described previously[85]. This distinguishes correct from incorrect peptide IDs in a manner analogous to the widely used Percolator algorithm (https://noble.gs.washington.edu/proj/percolator/), through employing a distinct machine learning algorithm. The following parameters were considered: XCorr, ΔCn, missed cleavages, peptide length, charge state, and precursor mass accuracy. Protein assembly was guided by principles of parsimony to produce the smallest set of proteins necessary to account for all observed peptides (algorithm described in[85]).

Proteins were quantified by summing TMT reporter ion counts across all matching peptide-spectral matches using 'MassPike', as described previously[84]. Briefly, a 0.003 Th window around the theoretical m/z of each reporter ion was scanned for ions, and the maximum intensity nearest to the theoretical m/z was used. The primary determinant of quantitation quality is the number of TMT reporter ions detected in each MS3 spectrum, which is directly proportional to the signal-to-noise (S:N) ratio observed for each ion. An isolation specificity filter with a cut-off of 50% was additionally employed to minimise peptide co-isolation[84]. Peptide-spectral matches with poor quality MS3 spectra (a combined S:N ratio of less than 250 across all TMT reporter ions) or no MS3 spectra at all were excluded from quantitation. Peptides meeting the stated criteria for reliable quantitation were then summed by parent protein, in effect weighting the contributions of individual peptides to the total protein signal based on their individual TMT reporter ion yields. Protein quantitation values were exported for further analysis in Excel.

For protein quantitation, reverse and contaminant proteins were removed, then each reporter ion channel was summed across all quantified proteins and normalised assuming equal protein loading across all channels. For further analysis and display in Figures, fractional TMT signals were used (i.e., reporting the fraction of maximal signal observed for each protein in each TMT channel, rather than the absolute normalised signal intensity). This effectively corrected for differences in the numbers of peptides observed per protein. *P* values were calculated using the method of significance A and corrected for multiple hypothesis testing in Perseus version 1.5.2.20.4[86].

**Immunoblot analysis of isolated PM fraction.** Following incubation with high-affinity streptavidin agarose beads as described above, beads were washed with lysis buffer, PBS/0.5% SDS and UC buffer, and incubated with 4x Laemmeli Sample buffer (Bio-Rad) and tris(2-carboxyethyl)phosphine powder (TCEP; Thermo Fisher Scientific) at 95 °C to elute proteins for immunoblot analysis.

**Immunoblotting**
Adipocytes were cultured in reduced media volumes as previously described for 48 h prior to the experiment (250 μL media per well of 24-well plate, media changed every 24 h) to improve insulin sensitivity[69]. Cells were washed twice in ice-cold PBS and lysed in RIPA lysis buffer (50 nM Tris pH 7.5, 150 mM NaCl, 1 mM EDTA, 1% Triton, 0.5% Na Deoxycholate, 0.1% SDS, 1% glycerol) containing EDTA-free Phosphatase and Protease Inhibitor Cocktail (Pierce). Cell lysates were sonicated and centrifuged at 16,000 x g, 20 min, 4 °C to remove lipid (top layer) and unlysed cells/debris (pellet). Protein concentration of the infranatant was quantified using a BCA Protein Assay Kit (Pierce) according to the manufacturer's instructions. Lysates were denatured by mixing with 4x Laemmeli Sample buffer, reduced with TCEP and heated at 65 °C for 10 min. Protein was resolved by SDS-PAGE using

4-20% (or 10% for C3ORF18) Mini-PROTEAN TGX Stain-Free Gels (Bio-Rad) and transferred to a nitrocellulose membrane using the Trans-Blot Turbo Transfer System (Bio-Rad). Membranes were blocked in 5% (w/v) dried skimmed milk (Marvel) in tris-buffered saline for 1 h at RT, followed by overnight incubation at 4 °C with the appropriate primary antibody (Supplementary Data 3). Membranes were incubated with the appropriate horseradish peroxidase (HRP) or fluorophore-conjugated secondary antibodies (Supplementary Data 3) diluted 1:5000 in 5% (w/v) dried skimmed milk for 1 h at RT. Signals were detected using enhanced chemiluminescence (ECL; Thermo Fisher Scientific) or in the appropriate fluorescence channel on a ChemiDoc MP Imaging System (Bio-Rad).

Band intensities were quantified using ImageLab (v6.1, Bio-Rad). Data in Fig. 5g and Supplementary Fig. 5j, v were min-max normalised within replicates to account for varying signal total intensity between replicates.

### PNGase F-mediated deglycosylation of N-linked glycans
Cells were lysed in RIPA buffer, and protein concentration was determined using a BCA assay. Proteins were reduced using 50 mM DTT, heated at 65 °C for 10 min and cooled on ice, and 1 μL/20 μg protein PNGase F (Promega) was added. Samples were incubated at 37 °C for 2 h before Laemmli buffer was added, and samples were resolved by SDS-PAGE.

### O-linked deglycosylation
O-linked deglycosylation was performed using the GlycoProfile™ β-Elimination Kit (Sigma, PP0540) according to the manufacturer's instructions. Briefly, glycoprotein samples were incubated with the β-elimination reagent mixture under alkaline conditions to chemically remove O-linked glycans, followed by neutralisation and cleanup using centrifugal filtration units. Deglycosylated protein samples were subsequently prepared for SDS-PAGE analysis.

### Isolation and subfractionation of primary rat adipocytes
This study used archived subcellular membrane fractions from epididymal fat pad tissue harvested from (n = 4) male Wistar rats (180–200 g), which were humanely killed using a UK Home Office Schedule 1 method. Ethical approval was granted by the University of Bath Animal Welfare and Ethical Review Body, and all animal procedures were conducted in accordance with UK Home Office regulations. Animals were housed under conditions of 12 h light:12 h darkness, a temperature of 21 ± 2 °C and relative humidity of 55 ± 10%. Standard chow diet and water were available ad libitum. For the purpose of preparing primary adipocytes from epididymal fat pads, only male rats could be used. Primary adipocytes were prepared from the whole epididymal fat pads according to the method previously described[87] with some modifications. The epididymal fat tissue was quickly removed and rinsed in 2.5% (w/v) BSA / Krebs-Ringers-HEPES buffer (KRH) (140 mM NaCl, 4.7 mM KCl, 2.5 mM CaCl₂, 1.25 mM MgCl₂, 2.5 mM NaH₂PO₄, 10 mM HEPES, pH 7.4) containing 200 nM adenosine (through the procedure the buffer is kept at 37 °C). The washed tissue was placed in KRH buffer (1 g of tissue / 2 mL) containing 1 mg/mL collagenase (Worthington, Type I), 3.5% (w/v) albumin, 5 mM glucose, and 200 nM adenosine and minced finely with scissors. The tissue suspension was shaken at 100 rpm in a shaking water bath at 37 °C for approximately 20 min until most of the tissue lumps were digested. The resulting cell suspension was filtered through a nylon mesh (250 μm mesh size, Lockertex), returned to 37 °C, and a fresh 2.5% (w/v) BSA/KRH buffer containing 200 nM adenosine was added, and the cells were allowed to float. The infranatant buffer was removed using a blunt needle (2 mm dia. x 100 mm) attached to a 20 mL plastic syringe, and 15–20 mL 2.5% (w/v) BSA/KRH buffer (with 200 nM adenosine) was added. The cells were gently resuspended and then allowed to float. This washing procedure was repeated 3 times. The cell suspension was adjusted to a cytocrit of 20% approximately, (using a capillary tube, which was centrifuged at 1500 g for 30 s, and expressed as the ratio of the length of the packed cell fraction in the tube to the total length of the suspension in the tube). The cells are then ready to be used. Insulin stimulation was performed with 20 nM insulin for 20 min. At the end of the stimulation, cells were washed once with KRH (no BSA supplementation) and once in HES buffer (20 mM HEPES, 1 mM EDTA, 250 mM sucrose, pH 7.4) supplemented with protease and phosphatase inhibitors and kept at 18 °C and homogenised with 10 strokes of a glass/Teflon Potter-Elvehjem homogeniser (Thomas Scientific, USA) before proceeding with the subcellular fractionation protocol.

Subcellular fractionation was performed by differential centrifugation in OptimaMax Ultracentrifuge according to the method of Simpson et al. (1983)[88]. Briefly, the homogenate was centrifuged at 1000 g for 1 min at 4 °C to separate fat from cellular fractions. The tubes were left on ice to solidify the fat layer. The homogenate was aspirated using a syringe attached to a 12 G hypodermic needle, and the pellet and fat layer on top were discarded. The homogenate was then spun at 17000 g for 20 min at 4 °C (TLA100.3 rotor). The pellet from this spin was resuspended and loaded on a sucrose cushion (20 mM HEPES, 1 mM EDTA, 1.12 M sucrose, pH 7.4) and spun at 104000 g for 20 min at 4 °C (TLS55 rotor). The plasma membrane fraction was collected from the interface and spun twice at 76000 g for 9 min at 4 °C (TLA 100.3 rotor) to remove remaining sucrose and contaminating endosomal fractions and cytosol. Purified plasma membranes were resuspended in 100 mM Tris, pH 8.0 + 2% SDS.

### siRNA knockdown in differentiated adipocytes
siRNA knockdown in 3T3-L1 adipocytes was performed as previously described[89,90]. Briefly, siRNA targeting mouse C3ORF18 (6430571L13Rik, siGENOME SMARTpool #235599; Dharmacon Horizon) at a final concentration of 50 nM was delivered to differentiated adipocytes (day 6 following differentiation) using TransIT-X2 (Mirus Bio). Cells were fed with fresh media 24 h following transfection, and transfected again 96 h after the initial transfection as previously described[90]. Functional assays were performed 96 h following the second transfection.

SGBS adipocytes were transfected as described above, with minor modifications. Forward transfection with Lipofectamine RNAiMax (Thermo Fisher Scientific) was used to deliver siGENOME SMART pool #51161 (human C3ORF18) at a final concentration of 50 nM on day 10 and day 14 after differentiation. Functional assays were performed 96 h following the second transfection.

**siRNA knockdown during differentiation.** 3T3-L1 fibroblasts were seeded on matrigel-coated 96-well Pheno plates (Revvity- 6055302) and grown for 3 to 4 days until 100% confluency. To maintain silencing of target genes throughout differentiation, forward transfection of siRNA was performed on days −2, 0, 3 and 6 of differentiation. Briefly, siRNA at final concentration 25 nM was delivered using Lipofectamine RNAimax (1:33, Thermo-13778075) diluted in OptiMEM. Mixture was incubated at room temperature for 20 min before added to cells in 3T3-L1 growth media described above. For Septin 2 knockdown, siGENOME SMARTpool #18000 was used.

### qPCR
RNA extractions were performed using the RNeasy Mini kit (Qiagen 1152 #74104). Concentrations of RNA samples were quantified using NanoDrop. cDNA synthesis from 500 ng RNA was performed using the GoScript Reverse Transcriptase kit (Promega #A2801). Real-time (RT)-polymerase chain reaction (PCR) was performed using TaqMan or

SYBR Green Master Mix on an ABI QuantStudio 5. Primers are described in Supplementary Data 3.

## Glucose transport

Glucose transport assays were performed as previously described[69]. Adipocytes were cultured in reduced media volumes as previously described for 48 h prior to the experiment (125 µl media per well of 48-well plate, media changed every 24 h) to improve insulin sensitivity[69]. Cells were serum-starved for 2 h in DMEM containing 0.2% BSA at 37 °C, 10% CO$_2$, washed and incubated in pre-warmed Krebs–Ringer phosphate (KRP) buffer containing 0.2% bovine serum albumin (KRP buffer; 0.6 mM Na$_2$HPO$_4$, 0.4 mM NaH$_2$PO$_4$, 120 mM NaCl, 6 mM KCl, 1 mM CaCl$_2$, 1.2 mM MgSO$_4$ and 12.5 mM HEPES (pH 7.4)) for 10 min, and then stimulated with 0.5 or 100 nM insulin for 20 min. To determine non-specific glucose uptake, 25 µM cytochalasin B (in ethanol, Sigma Aldrich) was added to control wells before the addition of 2-[$^3$H]deoxyglucose (2-DG) (PerkinElmer). During the final 5 min, 2-DG (0.25 µCi, 50 µM) was added to cells to measure steady-state rates of 2-DG uptake. Cells were then moved to ice, washed with ice-cold PBS, and solubilised in PBS containing 1% (v/v) Triton X-100. Tracer uptake was quantified by liquid scintillation counting on the TriCarb 2900TR (PerkinElmer), and data were normalised for protein content.

## Immunofluorescence

### Cell surface immunofluorescence plate assays (HA-C3ORF18/GLUT4/TFR)

PM translocation of HA-C3ORF18, GLUT4 and TfR in 3T3-L1 adipocytes was assessed as previously described[69,91], with minor modifications. Adipocytes were cultured in reduced media volumes as previously described for 48 h prior to the experiment (50 µl media per well of 96-well plate, media changed every 24 h) to increase insulin sensitivity[69]. Cells were serum-starved for 2 h and stimulated with insulin at doses and for times indicated. For assessment of PM translocation following EGF treatment, cells were serum starved for 2 h and treated with mouse EGF (Gibco, PMG8041) for 5 min at doses indicated. Where inhibitors were used, cells were pre-treated with 200 nM Wortmannin (Sigma, W1628), 10 µM GDC-0068 (Selleck Chemicals S2808), MK2206 (MedChemExpress, HY-10358) or U0126 (Cell guidance systems, S106) for 15 min prior to insulin stimulation. Cells were washed by gently immersing the 96-well plates 10 times in beakers containing ice-cold PBS (five washes in each of two 1 L beakers containing PBS; all subsequent PBS washes were performed using this method). Plates were placed on ice, and residual PBS was removed with a multi-channel pipette. Cells were fixed with 4% paraformaldehyde (PFA) for 5 min on ice and 10 min at room temperature. PFA was quenched with 50 mM glycine in PBS for 10 min. Cells were blocked with 5% normal swine serum (NSS; Dako, X0901) in PBS for 30 min. Cells were then incubated with primary antibody solution (Supplementary Data 3) in 2% NSS in PBS for 1 h at room temperature, washed in PBS, and incubated with secondary antibody solution in 2% NSS in PBS for 1 h at room temperature in the dark. Cells were stored in PBS containing 2.5% DABCO, 10% glycerol and pH 8.5, as well as sealed and kept at 4 °C in the dark before imaging. Plates were equilibrated to room temperature for 30 min before imaging.

Plates were imaged on the Perkin Elmer Opera Phenix High Content Screening System, and mid-section confocal images were obtained using a x20 water objective (NA 1.0), with 2-pixel binning. Excitation wavelengths and emission filters used were as follows: surface HA: 488 nm, 500–550 nm; endogenous surface TfR: 568 nm, 570–630; endogenous surface GLUT4: 647 nm, 650–760; and Hoechst: 405 nm, 435–480 nm. The microscope settings were automated and predefined before imaging the entire plate, meaning identical positions in each well were sampled across all wells and conditions without any human supervision. Nine positions towards the centre of each well

were selected for imaging. Following the acquisition, positions across each plate and well were inspected at random to ensure proper seeding/staining for quality assurance. Mean fluorescence intensity was calculated to determine the surface amount of protein.

Surface translocation of HA-C3ORF18 in L6 myoblasts and myotubes was performed as described above, with plates imaged using a x40 water objective with 40-50 fields of view per well used to quantify mean fluorescence intensity.

Surface translocation of HA-C3ORF18 in SGBS adipocytes was performed as described above with minor modifications. Cells were serum starved for 4 h, stimulated, washed with PBS and incubated with primary antibody on ice for 2 h. Cells were washed in PBS, fixed with 4% PFA for 5 min on ice and 10 min at room temperature and secondary antibodies were added as described above. Plates were imaged using a x40 water objective, and 18 fields of view were used to quantify mean fluorescence intensity in each well.

### Colocalisation immunofluorescence (Opera Phenix)

Immunofluorescence to assess colocalisation of C3ORF18, GLUT4 and TfR was performed as described above with 0.1% saponin added to the blocking, primary and secondary antibody solutions to permeabilise cells. Plates were imaged on the Perkin Elmer Opera Phenix High Content Screening System, and mid-section confocal images were obtained using a ×63 water objective. Colocalisation Threshold plugin in ImageJ Fiji v2.9.0 was used to compute Mander's coefficient.

### Immunofluorescence (SP8)

SEPTIN-2 and SEPTIN-11:

Differentiated 3T3-L1 adipocytes were reseeded onto glass coverslips, fixed with 4% PFA in PHEM buffer (60 mM PIPES, 25 mM HEPES, 10 mM EGTA, 10 mM MgCl$_2$ pH 6.8), permeabilised with 0.1% saponin in PHEM buffer and blocked with 5 % NSS in PBS. Samples were incubated with primary antibodies (Supplementary Data 3) in 2% NSS, 0.1% saponin in PBS for 1 h at room temperature. Coverslips were washed with 0.1% saponin in PBS and incubated with secondary antibodies and BODIPY™ 493/503 (1:1000, Invitrogen, D3922) in 2% NSS in PBS for 1 h at room temperature. Imaging was performed using a Leica SP8 microscope with a x63 Plan-Apochromat oil objective. Image acquisition was performed using Leica Application Software (Leica, v3.5.7.2) and images noise reduced using the median module with 2 iterations.

C3ORF18:

Images acquired on Leica SP8 (Fig. 4d, Supplementary Fig. 4i) were further deconvoluted using the Deconvolution Express module in Huygens Professional 25.10 using default settings. Confocal planes identified as having the best focus are used for representation in the manuscript.

## Quantitative assessment of lipid droplet accumulation during adipogenesis

Longitudinal quantification of lipid droplet accumulation during adipogenesis was performed with the Digital Phase Contrast (DPC) imaging module of the Perkin Elmer Opera Phoenix High Content Screening system. Object thresholding and mean intensity analysis measurements were performed using Harmony 5.2 analysis software. On day 8, samples were fixed and stained with Hoechst to determine the total cell number.

## Statistical analysis

All statistical analyses were performed in Graphpad Prism 10 unless otherwise stated. Two-tailed paired Student's tests were used to compare the means between 2 groups. One/two-way ANOVA with Šidák correction for multiple comparisons was used for multigroup comparisons. A mixed-effects model was used for multiple comparisons in Supplementary Fig. 5g, Supplementary Fig. 5j and v. Variations among replicates were expected to have normal distributions and equal variance.

## Reporting summary

Further information on research design is available in the Nature Portfolio Reporting Summary linked to this article.

## Data availability

The proteomics data generated in this study have been deposited in the PRIDE database under the accession code PXD061017 for the LOPIT-DC data and PXD061616 for the plasma membrane proteomics data. Source data are provided with this paper.

## Code availability

No new code was developed for this study. LOPIT-DC code is publicly available at https://github.com/CambridgeCentreForProteomics/adipocyteLOPIT2025.

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

## Acknowledgements

We thank Professor Martin Wabitsch (Division of Paediatric Endocrinology and Diabetes, University Medical Centre Ulm, Germany) for generously providing SGBS cells and Professor David James (University of Sydney) for kindly providing 3T3-L1 and L6 cells. We also thank Mike Deery (Cambridge Centre for Proteomics) for performing the LOPIT mass spectrometry analysis and Yagnesh Umarania (Cambridge Centre for Proteomics) for assistance with LOPIT proteomic data processing. We thank Robin Antrobus (Cambridge Institute for Medical Research Proteomics Facility) for plasma membrane mass spectrometry analysis. This work was supported by the Cell Imaging Facility at the Institute of Metabolic Science, who were funded by the Medical Research Council [MC_UU_00039]. This work was supported by an MRC Career Development award (MR/S007091/1) and a Project grant (MR/Z504592/1) awarded to D.J.F. O.J.C. was supported by a Wellcome Trust PhD studentship. J.A.C. was supported by a BBSRC iCASE award with AstraZeneca (BB/R505304/1). L.M.B. was supported by the EU Horizon 2020 programme INFRAIA project EPIC-XS (project 823839). D.C.G. was supported by a Biotechnology and Biological Sciences Research Council project grant (BB/W005905/1) and a Wellcome Trust/Royal Society Sir Henry Dale Fellowship (210481). D.B.S. is supported by the Wellcome Trust (WT 219417), the MRC (MR/X00970X/1), and the National Institute for Health Research (NIHR) Cambridge Biomedical Research Centre and NIHR Rare Disease Translational Research Collaboration. MPW was supported by a Wellcome Trust Discovery Award 309425/Z/24/Z and a Medical Research Council Project Grant (MR/X000516/1).

## Author contributions

O.J.C. and D.J.F. conceived and led the project. O.J.C., J.A.C., D.J.F., and K.S.L. designed LOPIT-DC proteomic experiments, O.J.C. performed the LOPIT-DC fractionation, and L.M.B. and O.J.C. performed analyses of LOPIT-DC data. O.J.C., D.J.F., and M.P.W. designed plasma membrane proteomics experiments, O.J.C. performed the plasma membrane biotinylation, O.J.C. and H.L performed sample preparation, and H.L. performed data analyses. O.J.C. generated C3ORF18 cell lines. O.J.C. and D.M. performed analysis of C3ORF18 localisation. O.J.C., M.B., D.M., B.L.H., and D.J.F. performed siRNA-mediated knockdown experiments and assessment of insulin responses by immunofluorescence, glucose transport assays and Western blotting. L.L. performed Western blotting, and D.M. assessed septin localisation by immunofluorescence. D.M. and B.L.H. performed experiments assessing the effect of septin and C3ORF18 knockdown on adipocyte differentiation. M.B. generated C3ORF18 deletion mutants and performed corresponding Western blot analyses for molecular weight shift. M.B. performed Western blotting of C3ORF18 protein abundance over adipocyte differentiation. F.K. performed analyses in primary rodent adipocytes. S.P. designed the lipid droplet isolation and contributed to discussions on C3ORF18. D.C.G. contributed to discussions on endosomal trafficking and the role of C3ORF18. D.B.S. contributed to discussions. O.J.C. and D.J.F. draughted the manuscript, and all authors contributed to the final version.

## Competing interests

The authors declare no competing interests.
