## [Transparent Peer Review file · Nature Communications]

Dynamic subcellular proteomics identifies regulators of adipocyte insulin action

Corresponding Author: Dr Daniel Fazakerley

Version 0:

Reviewer comments:

Reviewer #1

(Remarks to the Author)

Conway et al report an analysis of the proteome of adipocytes incubated with and without insulin, and a qualitative analysis of the changes in the adipocyte plasma membrane proteome in response to insulin. Overall, this work is well performed, interesting and will, as the authors note, be a useful resource for the field.

However, on balance I found this work to be quite descriptive in places, I have some concerns with the data analysis and overall found this to be a limited mechanistic advancement.

Specific issues:

Of the proteins which redistribute in response to insulin, how many of these are specific for insulin? This is an important question which is not addressed. It is known that (for example) endosomal trafficking and cell signaling can be influenced by a wide array of growth factors and receptors, so a difficulty I have with this data is the assertion that the proteins which move are targets of the insulin cascade. I appreciate this is a huge ask, and would not suggest that the authors need to do experimental work to address this, but this places a rather large caveat over the data and its interpretation, which the authors do not address. This could be addressed by some of the comments on Figure 4.

Figure 4a is not convincing. Figure 4b should display controls and comparison with GLUT4. The data in figure 4c relate to my point above. The effect of insulin is small, ~70%. This is of a magnitude observed for endosomal proteins, like the transferrin receptor. Is this insulin-specific? This could be an example whereby the authors address the issues noted above.

Figure 5. I would like to see the authors also present this data so that 5a-5d are normalized to the control (0 insulin) for each data set. The knockdown data are on average lower for all parameters. These are not statistically significant, but a lower basal value means that the fold increase in 2-DG, GLUT4 levels, etc., may not be as significantly different as the absolute values show. When analyzed this way, is there a difference between control and knockdown? There may well be, but this should be presented. Are GLUT4 levels changed when C3ORF18 is knocked-down?

Figure 1c reveals that the plasma membrane is enriched in fraction 8, with little evidence of markers in fraction 7 (pink line in 1c). Why does insulin stimulation not enrich GLUT4 and IRAP in fraction 8 (figure 2b)?

Minor issues:

Figure 2D is not discussed in the text.

This reviewer did not understand what should be taken from Figure 3A; could you improve the legend?

Reviewer #2

(Remarks to the Author)

This manuscript entitled "Dynamic subcellular proteomics identifies novel regulators of adipocyte insulin action" presents a comprehensive study of insulin-regulated protein redistribution in 3T3-L1 adipocytes using advanced subcellular proteomics

approaches, including LOPIT-DC and surface biotinylation. The findings are exciting and likely to be of broad interest. The authors provide a detailed map of the subcellular localization of the adipocyte proteome, where many proteins align well with their expected localizations. The strength of the approach is further supported by the observation of known insulin-induced protein relocalizations and phosphorylation events.

Notably, the identification of a novel candidate regulator, C3ORF18, as an insulin-responsive protein is intriguing and may open new avenues for research in insulin signaling and trafficking.

I commend the authors for their efforts to validate the identity of C3ORF18, a protein with a predicted molecular weight (MW) of ~17 kDa, observed as a ~40 kDa species. However, Figure 4a is not entirely convincing. While there is an increase in signal intensity upon insulin stimulation, the smear-like appearance of the signal lacks a clear, discrete band, which complicates interpretation. Given that C3ORF18 is barely detectable under basal conditions (as shown in Figure 4a and Supplementary Figure 4c), it is unclear how the authors conclude that it undergoes subcellular relocalization.

Furthermore, there appears to be an inconsistency between Figures 4a and 4b: C3ORF18 is not expressed in 3T3-L1 cells at day 0 in Figure 4a, yet it is detected in siControl cells in Figure 4b. Could the authors clarify this discrepancy? Is there a corresponding change in C3ORF18 MS intensity between basal and insulin-stimulated conditions?

Additionally, the nature of the apparent MW shift observed for C3ORF18 should be addressed. What post-translational modifications (e.g., glycosylation) could account for this substantial shift on SDS-PAGE? It would be useful to treat the samples with glycosidases to determine whether the migration pattern changes, supporting the presence of glycosylation or other modifications.

To strengthen Figure 4b, I suggest including DAPI staining to help assess whether C3ORF18 localizes to the nucleus or perinuclear region. In the current images, the distinction is unclear, and co-staining would provide valuable spatial context. Moreover, in basal conditions, the apparent association of C3ORF18 with ribosomes (Figure 3d) suggests a possible localization to the rough ER, which aligns with its predicted membrane-associated nature. Co-localization experiments with ER markers, or proximity ligation assays (PLA), would help confirm this interpretation.

The observed relocalization of C3ORF18 following inhibition of PI3K and AKT appears robust; however, a representative image would help the reader better appreciate the morphological differences under these conditions.

As an additional analysis, I recommend exploring C3ORF18 dependency using large-scale CRISPR screen datasets, filtered for adipocyte or metabolic cell types. This could offer insights into functional interactions or pathways related to C3ORF18.

Lastly, the term "PM" (plasma membrane) is used in the abstract without being defined. Please define it upon first use.

This study presents valuable insights into the spatial dynamics of the adipocyte proteome in response to insulin. While the findings regarding C3ORF18 are compelling, several points require clarification or additional validation to solidify its proposed role.

Reviewer #3

(Remarks to the Author)

In their study, Conway et al. use LOPIT, a subcellular proteomics approach, in a 3T3-L1 mouse cell model to screen for protein relocalizations following insulin stimulation. They identify C3ORF18, a protein of unknown function, as a potential novel player in insulin signaling. The manuscript is clearly structured, and the data are clearly presented and accessible via a Shiny app. The approach is interesting and allows for the unbiased identification of novel components involved in signaling.

However, the biological relevance of the study is somewhat limited, and the overall outcome modest. The new insights gained are relatively minor, and the follow-up analysis of C3ORF18 is very superficial. The function of the protein remains unclear, and its role in insulin signaling is not convincingly established. In addition, the information about the used methods in the text itself is too short and difficult to follow, lacking the necessary detail to fully understand the data.

Overall, while the technical approach is of interest, the study would benefit from a more thorough characterization of C3ORF18 and clearer methodological documentation. As it stands, the manuscript presents only limited biological insight. Specific points:

- Please provide a clearer explanation in the main text of how the LOPIT approach works, including the principle of fractionation, and how BUNDLE assigns proteins to compartments based on distribution profiles. It is currently unclear how marker sets are chosen and whether they are assumed to remain static between conditions.
- The number of markers used appears large, and it seems they are not reassigned under insulin stimulation. Does the method assume these markers are static and do not relocalize? If so, this is a potential limitation.
- While some organelle profiles (e.g., mitochondria) are clearly separated, others, such as plasma membrane and Golgi, appear overlapping and are harder to distinguish. This is also reflected in their lower concordance with published data. Please provide a quantitative metric or quality score for the compartment assignments to support interpretability.
- Line 111: If proteins in the ER–Golgi–PM axis show low concordance with other datasets, how can changes in localization be confidently attributed to insulin signaling rather than inherent trafficking variability?
- From ~4000 detected proteins, only ~1700 were assigned to compartments. Please explain why the remaining proteins

could not be assigned. What thresholds or criteria (e.g., probability cutoffs) were used and how was it set? Are proteins with lower assignment confidence also more likely to show differential localization?

- Please describe in more detail in the text what the second proteomics method is. Can you compare the changes in the PM detected via LOPIT-DC and biotinylating? What is different and why? Why was it necessary in addition to LOPIT? What new information does it provide that LOPIT does not?
- What happens to lipid droplets in the P1–P8 fractions? Could LD-associated proteins be counted more than once due to their presence in both the LD fraction and earlier fractions?
- Septins are discussed in the context of lipid droplets. Do any septins relocalize in response to insulin? What is their possible role in adipocyte biology and lipid storage?
- Interpretation of ER accumulation upon insulin: Some proteins show ER accumulation upon insulin treatment. How do you interpret this? Could this indicate a block or re-routing in vesicular trafficking rather than a classical translocation?
- Protein abundance vs. localization change (Line 138): For IRAP, the differential localization probability is high, but the shift appears to reflect abundance rather than spatial redistribution. How does your model distinguish these scenarios?
- Line 146: What is the exact diff. loc. prob. threshold used to define significant relocalization? For proteins like mTOR, the localization shift is not obvious from the profiles—please clarify.
- Line 153: When proteins like DEPDC5 or SZT2 are said to dissociate from lysosomes, where do they relocalize? Is this to the cytosol or another compartment?
- Line 195: Has C3ORF18 been identified in other proteomic studies of adipocyte differentiation? If so, please cite relevant references.
- Figures 4b and 4e: The nuclear localization of C3ORF18 is suggested, but no nuclear marker is shown. Likewise, endosomal markers should be used to support the claim that the protein localizes to insulin-responsive endosomes.
- Line 242: The effect of C3ORF18 knockdown on AKT signaling is claimed, but no clear quantification is shown, and the Western blot quality is suboptimal. Please provide quantitative data and clearer blots.
- Clarify C3ORF18 relocalization dynamics: The current description is ambiguous. Does C3ORF18 move from the nucleus, endosomes, or another compartment to the plasma membrane? How do endosomes fit into the proposed mechanism?
- Predicted interactors of C3ORF18: Many predicted interactors are mitochondrial proteins. How does this reconcile with its observed localization and proposed function in insulin signaling?
- Line 248: The statement that C3ORF18 improves insulin sensitivity in adipose tissue is not supported by the presented data. This conclusion should be supported with additional experiments.
- The functional role of C3ORF18 doesn't become clear. Is its relocalization upon insulin treatment dependent on phosphorylation? Is its role primarily in signaling or trafficking? Based on your data, how might C3ORF18 mechanistically influence the insulin signaling cascade? A speculative model or hypothesis would strengthen the discussion.

Minor comments:

- Line 43: phrasing should be improved.
- Line 123 and 126: Extended data Fig.1h is mentioned before Fig.1g.
- Please change colours of PM markers (basal vs insulin) in Fig. 3e so those can be distinguished by the reader.
- Any idea on what the band at 80-90kDa is in the C3ORF18 blots?
- Line 204: there is no figure 3f.

Version 1:

Reviewer comments:

Reviewer #1

(Remarks to the Author)

The authors have made a careful, balanced and nuanced response to all of the points I raised. While I still remain a little 'on the fence' about the overall impact, the revised paper is a strong contribution to the field and therefore I am happy to indicate that in my opinion, the paper is now worthy of publication.

Reviewer #2

(Remarks to the Author)

The authors provided thorough and thoughtful responses to previous comments, and made substantial revisions to the manuscript. Overall, I believe the revised version represents a clear improvement.

In particular, the authors have convincingly addressed my concerns regarding the biochemical identity of C3ORF18 and the apparent discrepancy between its predicted and observed molecular weight. The additional biochemical analyses (including glycosylation, lysine modification, phosphorylation, dimerization, and intrinsic disorder assessments) provide a comprehensive and convincing explanation for the altered electrophoretic mobility and significantly strengthen confidence in the specificity of the signal.

The expanded high-resolution imaging and colocalization analyses substantially improve the interpretation of C3ORF18 subcellular localization. The use of multiple compartment markers, together with quantitative colocalization analyses, now clearly supports the conclusion that C3ORF18 resides predominantly in post-Golgi/endosomal compartments, with partial overlap with TfR-positive structures and limited overlap with GLUT4 and ER markers. The clarification that C3ORF18 localizes to a perinuclear region rather than the nucleus is also helpful.

While the precise molecular function of C3ORF18 remains unresolved, I think the authors now provide a well-supported model that is appropriate for the scope of the study and represent a significant characterization of this protein.

Reviewer #3

(Remarks to the Author)

The authors have clearly invested significant effort in addressing the points raised during review. The technical concerns have been handled carefully, the LOPIT/BUNDLE analysis is now much clearer, and the addition of orthogonal validation strengthens confidence in the dataset. Importantly, the revised manuscript avoids overstatement, and the limitations of the approach are now explicitly and appropriately discussed. These changes substantially improve the readability and credibility of the work.

That said, while the new data provide additional insight into C3ORF18, a comprehensive mechanistic understanding is still missing. The authors convincingly demonstrate its localization, dependence on AKT signaling, and the consequences of C3ORF18 loss on TfR, GLUTs, IRS1, and insulin responsiveness. However, there is still no direct evidence for the underlying molecular mechanism, such as defined binding partners, involvement of specific trafficking machinery, or biochemical activity. In addition, the finding that C3ORF18 also responds to EGF introduces a conceptual shift, suggesting that it functions downstream of shared PI3K/AKT signaling rather than as an insulin-specific regulator.

Overall, I appreciate the authors' transparent presentation of the limitations of their study. While the work does not yet provide a complete mechanistic picture, it represents a solid and carefully executed contribution that will be useful to the field.

Response to Reviewers

We thank all reviewers for their time and feedback on our manuscript. In response, we have substantially revised our manuscript, most notably:

1. Provided greater detail and clarity surrounding data analysis and interpretation of the subcellular proteomics (LOPIT) data.
2. Performed higher resolution imaging of C3ORF18 localisation. These new data provide more clarity of C3ORF18 subcellular localisation and suggest that C3ORF18 resides, at least partially, in a distinct vesicular compartment to GLUT4 and TfR.
3. Provided a more detailed biochemical assessment of C3ORF18, including a possible explanation for its higher apparent molecular weight on SDS-PAGE.
4. Included data on C3ORF18 translocation in primary adipocytes.
5. Included additional data on how loss of C3ORF18 impairs adipocyte responses through altered protein abundance of insulin signalling intermediates and glucose transporters.

We have also included more minor additions, including:

1. Additional imaging of Septin localisation in human adipocytes.
2. New analysis of the effect of *Sept2* knockdown on adipocyte differentiation.
3. Assessment of C3ORF18 translocation in response to an alternate growth factor, EGF, showing that this protein is not uniquely responsive to insulin.

We hope you and the Reviewers agree that these changes have substantially improved the accessibility, scope and impact of our manuscript. Please see point-by-point responses to individual reviewer comments below (in blue), with specific reference to changes to the manuscript text in red.

Reviewer 1

Conway et al report an analysis of the proteome of adipocytes incubated with and without insulin, and a qualitative analysis of the changes in the adipocyte plasma membrane proteome in response to insulin. Overall, this work is well performed, interesting and will, as the authors note, be a useful resource for the field.

However, on balance I found this work to be quite descriptive in places, I have some concerns with the data analysis and overall found this to be a limited mechanistic advancement.

We have addressed concerns around data analysis in detail below and edited the manuscript accordingly. We acknowledge that there are some more descriptive sections within the manuscript, but suggest that these are necessary to some extent when introducing a new large data set. We have now added additional data that

provides more mechanistic insight, described below.

Specific issues:

Of the proteins which redistribute in response to insulin, how many of these are specific for insulin? This is an important question which is not addressed. It is known that (for example) endosomal trafficking and cell signaling can be influenced by a wide array of growth factors and receptors, so a difficulty I have with this data is the assertion that the proteins which move are targets of the insulin cascade. I appreciate this is a huge ask, and would not suggest that the authors need to do experimental work to address this, but this places a rather large caveat over the data and its interpretation, which the authors do not address. This could be addressed by some of the comments on Figure 4.

Our studies have focused on protein translocation in response to insulin in adipocytes. We have not specifically studied whether proteins we found to redistribute with insulin also moved with alternate growth factors. Many growth factor RTKs utilise similar intracellular signalling machinery (e.g., PI3K/AKT, MAPK), so it is entirely possible that insulin responsive proteins are responsive to alternate growth factors. However, and as stated by the Reviewer, undertaking a comprehensive proteomics profiling of adipocytes/different cell types in response to multiple growth factors is beyond the scope of this study.

We have performed experiments in our adipocyte model to, at least partially, address this question. 3T3-L1 adipocytes express EGFR and increase glucose uptake in response to EGF (PMID: 7917789) (**Reviewer Figure 1A**). As with insulin, EGF-stimulation increased phosphorylation of AKT and MAPK (**Reviewer Figure 1B**), albeit with more transient kinetics than insulin. EGF-mediated AKT phosphorylation was highest at 5 min, but returned towards baseline by 15 min (**Reviewer Figure 1B**). With this in mind, we assessed whether 5 min EGF stimulation caused plasma membrane translocation of GLUT4, TfR and C3ORF18 (**New Extended Data Figure 4I**). GLUT4, TfR and C3ORF18 increased plasma membrane abundance in response to 5 min EGF stimulation, suggesting that C3ORF18 (and indeed GLUT4 and TfR) translocation is likely driven by AKT activation and is not specific to insulin signalling in adipocytes.

See Page 6: As for GLUT4 and TfR, HA-C3ORF18 translocation was not unique to insulin, as 5 min EGF-stimulation also increased PM abundance (Extended Data Fig. 4I).

Reviewer Figure 1: a) Glucose uptake in 3T3-L1 adipocytes following 5 min (left) or 15 min (right) stimulation with EGF or insulin at indicated doses. b) Western blot of phosphorylated AKT (Thr308 and Ser473) and TBC1D4 (Thr642) in 3T3-L1 adipocytes following EGF or insulin stimulation at indicated doses for 5 (top) or 15 (bottom) min. GAPDH* loading control for pTBC1D4, pAKT308 and pERK, GAPDH** loading control for pAKT473. One-way ANOVA with Dunnett's post-hoc test, * $p < 0.05$; ** $p < 0.01$.

Figure 4a is not convincing.

The commercial C3ORF18 antibodies currently available are suboptimal for Western blotting. Nevertheless, we have optimised our workflow (using 4-20% gradient gels, increased concentration of primary antibody to 1/500) to generate a better, more convincing blot, clearly showing an increase in C3ORF18 protein during adipocyte differentiation (**New Fig. 4a**). We have also included corresponding qPCR data, showing increased *C3orf18* mRNA over 3T3-L1 differentiation (**New Fig. 4b**).

Figure 4b should display controls and comparison with GLUT4.

We have revised this figure to include GLUT4 (**New Fig. 4c**). We are not sure which additional controls the Reviewer is suggesting.

The data in figure 4c relate to my point above. The effect of insulin is small, ~70%. This is of a magnitude observed for endosomal proteins, like the transferrin receptor. Is this insulin-specific? This could be an example whereby the authors address the issues noted above.

We agree with the Reviewer that the magnitude of the HA-C3ORF18 translocation response in original Figure 4C is consistent with the magnitude observed for endosomal proteins like TfR. Indeed, this is also consistent with the kinetics of HA-C3ORF18 translocation more closely matching TfR, than GLUT4, and the colocalisation of C3ORF18 with TfR-positive structures (**New Figure 4d-i**). Please also see our response to Reviewer 2 below for further discussion of C3ORF18 localisation.

Regarding the specificity of the response to insulin or other growth factors. We have studied whether EGF-stimulation invokes GLUT4 and/or C3ORF18 translocation. These data are described above (**Reviewer Figure 1**) and included in the revised manuscript (**New Extended Data Figure 4l**). These data suggest that the C3ORF18 translocation response is not specific to insulin.

Figure 5. I would like to see the authors also present this data so that 5a-5d are normalized to the control (0 insulin) for each data set. The knockdown data are on average lower for all parameters. These are not statistically significant, but a lower basal value means that the fold increase in 2-DG, GLUT4 levels, etc., may not be as significantly different as the absolute values show. When analyzed this way, is there a difference between control and knockdown? There may well be, but this should be presented. Are GLUT4 levels changed when C3ORF18 is knocked-down?

We understand and agree with the reviewers' points. In short, we have now included data in the revised manuscript that *C3ORF18* knockdown lowers GLUT1 and GLUT4 abundance (**New Figure 5f**). This likely explains the slightly lower basal 2-DG in knockdown cells. We also observe a substantial reduction in insulin signalling to AKT and TBC1D4 (**New Figure 5g, New Extended Data Figure 5j**). Therefore, lower overall insulin-stimulated glucose transport and GLUT4 translocation is likely due to the combined lower glucose transporter expression and insulin signalling responses, especially at lower more physiological insulin doses.

As the Reviewer requested, we have plotted the 2-DG uptake data as fold-change relative to basal as the reviewer requested (**Reviewer Figure 2A-D**). We have also plotted the surface TfR and GLUT4 data in this way. For completeness, we have also plotted the absolute change in 2-DG uptake at each dose of insulin (**Reviewer Figure 2E-H**). In almost all cases, with the exception of SGBS 2DOG uptake, the fold change data show a significant or a trend towards reduced responses with *C3orf18* knockdown. However, these fold-change data are highly variable as the basal values are very low and close to the limit of detection in our assays, and a small change in the basal signal has a large effect on the fold change. This is especially the case for 2-DG uptake data (and especially in SGBS assays) where basal DPM values are close to the background counts. The results of these analyses are summarised below:

3T3-L1 2-DG data: These data show no effect on fold-change, although a trend towards reduced fold changes with *C3orf18* knockdown, especially at 100 nM insulin. Absolute changes in 2-DG with insulin were impaired in knockdown cells. Although these fold-change may indicate a normal insulin response, please note the very high variability between experiments, largely due to low basal counts.

SGBS 2-DG data: These data show no effect on fold-change, but impaired absolute change in 2-DG. Although these fold-change may indicate a normal insulin response,

please note the very high variability in the *C3ORF18* KD condition, due to very low basal counts.

3T3-L1 GLUT4 data: These data show impaired fold-change and absolute change in surface GLUT4 with *C3orf18* knockdown.

3T3-L1 TfR data: These data show impaired fold-change and absolute change in surface TfR with *C3orf18* knockdown.

Overall, although noisy due to low signal in basal samples, these fold-change data indicate that total expression of glucose transporters and/or TfR are likely contributing to the overall lower responses observed with *C3ORF18* knockdown, as the reviewer suspected. Since we are now including additional data and discussion on GLUT4 abundance, we have decided to keep the raw data presented in the manuscript (**Figure 5a-d**) and not include the fold-change data (which is supplied below for the reviewers).

Please see below for our updated description of these data in the text on page 7:

Decreased insulin-stimulated 2-deoxyglucose transport after 8 d *C3orf18* knockdown was likely driven by both lower protein abundance of the glucose transporters GLUT4 and GLUT1, independent of mRNA expression (Fig. 5f, Extended Data Fig. 5i), as well as reduced proximal insulin signalling (Fig. 5g, Extended Data Fig. 5j). Lower insulin signalling responses were independent of changes in INSR mRNA and protein abundance or localisation (Extended Data Fig. 5k-m), but were associated with decreased IRS1 protein, as well as reduced expression of adipocyte marker genes (Extended Data Fig. 5n-p).

Reviewer Figure 2: Fold change in fluorescence intensity of surface GLUT4 (a) and TfR (b) in control and *C3orf18*-depleted 3T3-L1 adipocytes stimulated with 0.5 or 100 nM insulin for 30 min compared to unstimulated (basal) cells ($n = 6$ biological replicates).

Fold change in 2-deoxyglucose (2-DG) uptake in response to insulin in 3T3-L1 (c) and SGBS (d) adipocytes following C3ORF18 depletion (n = 8 and 5 biological replicates in 3T3-L1 and SGBS adipocytes, respectively). Absolute increases in fluorescence intensity of surface GLUT4 (e) and TfR (f) in control and C3orf18-depleted 3T3-L1 adipocytes stimulated with 0.5 or 100 nM insulin for 30 min compared to unstimulated (basal) cells (n = 6 biological replicates). Absolute increases in 2-deoxyglucose (2-DOG) uptake in response to insulin in 3T3-L1 (g) and SGBS (h) adipocytes following C3ORF18 depletion (n = 8 and 5 biological replicates in 3T3-L1 and SGBS adipocytes, respectively). All data represented as mean ± SEM. n.s. non-significant; *p < 0.05; **p < 0.01; ***p < 0.001; ****p < 0.0001 by two-way ANOVA and Holm-Šídák post-hoc tests.

Figure 1c reveals that the plasma membrane is enriched in fraction 8, with little evidence of markers in fraction 7 (pink line in 1c). Why does insulin stimulation not enrich GLUT4 and IRAP in fraction 8 (figure 2b)?

Figure 1c shows that the proteasome (purple) is enriched in fraction 8, not the plasma membrane (pink). The plasma membrane is more enriched in fraction 4, with some minor enrichment in fractions 1 and 2. The GLUT4 and IRAP profiles, for example in Figure 2b, show leftward shift in profile with insulin, from fraction 8 towards the earlier fractions. This is consistent with movement to the plasma membrane – although we cannot conclude this from the LOPIT profile data alone.

Minor issues:

Figure 2D is not discussed in the text.

Figure 2d is referred to in the results: “Differentially localised proteins were enriched for signalling terms (Fig. 2c), including insulin signalling, and for proteins with insulin-regulated phosphosites (Fig. 2d, Supplementary Table 1).”

The legend for Figure 2d is:

“Enrichment of proteins with insulin-regulated phosphorylation sites in 3T3-L1 adipocytes as reported by Fazakerley (2023)¹⁰ or Humphrey (2013)¹ in all proteins vs. high-confidence differentially localised proteins (*diff. loc. prob.* = 1). Fisher’s exact test, Fazakerley $p = 2.2 \times 10^{-11}$, Humphrey $p = 1.3 \times 10^{-13}$.”

We have added additional text to more clearly emphasise the data in 2d. See page 4:

Differentially localised proteins were enriched for signalling terms (Fig. 2c), including insulin signalling, and for proteins previously identified to contain insulin-regulated phosphosites in 3T3-L1 adipocytes (Fig. 2d, Supplementary Table 1).

This reviewer did not understand what should be taken from Figure 3A; could you improve the legend?

Thank you for pointing this out. We have now improved the legend as requested for Fig. 3a for greater clarity:

a) Chord plot visualising movement of proteins predicted to relocalise with high-confidence (*diff. loc. prob.* = 1) assigned to an organelle in at least 1 condition. The coloured lines indicate movement between compartments, with the arrowhead indicating direction of movement. For example, the two dark green blocks emanating from the “Golgi Apparatus” indicate movement of proteins from the Golgi to “Undefined” or the “Plasma Membrane”.

Reviewer #2 (Remarks to the Author):

This manuscript entitled “Dynamic subcellular proteomics identifies novel regulators of adipocyte insulin action” presents a comprehensive study of insulin-regulated protein redistribution in 3T3-L1 adipocytes using advanced subcellular proteomics approaches, including LOPIT-DC and surface biotinylation. The findings are exciting and likely to be of broad interest. The authors provide a detailed map of the subcellular localization of the adipocyte proteome, where many proteins align well with their expected localizations. The strength of the approach is further supported by the observation of known insulin-induced protein relocalizations and phosphorylation events.

Notably, the identification of a novel candidate regulator, C3ORF18, as an insulin-responsive protein is intriguing and may open new avenues for research in insulin signaling and trafficking.

Thank you very much for your positive assessment of our study.

I commend the authors for their efforts to validate the identity of C3ORF18, a protein with a predicted molecular weight (MW) of ~17 kDa, observed as a ~40 kDa species. However, Figure 4a is not entirely convincing. While there is an increase in signal intensity upon insulin stimulation, the smear-like appearance of the signal lacks a clear, discrete band, which complicates interpretation. Given that C3ORF18 is barely detectable under basal conditions (as shown in Figure 4a and Supplementary Figure 4c), it is unclear how the authors conclude that it undergoes subcellular relocalization.

We understand the concern regarding the ‘smear-like’ band for C3ORF18. We have generated a better, more convincing blot, clearly showing an increase in C3ORF18 protein during adipocyte differentiation (**New Fig. 4a**). We have also included corresponding qPCR data indicating increased *C3orf18* expression over adipocyte differentiation (**New Fig. 4b**).

We apologise for the confusion caused between Fig 4a and Extended Data Fig. 4c.

- Fig. 4a showed increased expression of C3ORF18 over differentiation of 3T3-L1 adipocytes. Here we observe very low protein abundance in preadipocytes, with increasing expression during differentiation.
- Extended Data Fig. 4c showed the plasma membrane fraction of differentiated 3T3-L1 adipocytes (day 10) which were untreated (basal) or treated with 100 nM insulin for 20 min. These data show that C3ORF18 abundance increased in the plasma membrane fraction of differentiated 3T3-L1 adipocytes following insulin treatment.

We have updated the legend of Extended Data Fig. 4c:

Western blot analysis of C3ORF18 in the affinity-purified PM fraction of differentiated (day 10) 3T3-L1 adipocytes isolated for plasma membrane proteomics as described in Fig. 3b.

Furthermore, there appears to be an inconsistency between Figures 4a and 4b: C3ORF18 is not expressed in 3T3-L1 cells at day 0 in Figure 4a, yet it is detected in siControl cells in Figure 4b. Could the authors clarify this discrepancy? Is there a corresponding change in C3ORF18 MS intensity between basal and insulin-stimulated conditions?

We again apologise that this was not clear from the Figure legends.

The data in Fig 4a show increased expression of C3ORF18 over differentiation of 3T3-L1 adipocytes. At day 0, these cells are pre-adipocytes, where we could not detect the C3ORF18 protein.

The data in 4b are micrographs of differentiated adipocytes (corresponding to day 10 in Fig 4a). Here we are showing the localisation of endogenous C3ORF18 in basal and insulin-stimulated cells.

As Fig. 4b does not have siRNA knockdown conditions – the Reviewer may be referring to Extended Data Fig. 4b. This Western blot shows the abundance of C3ORF18 in differentiated adipocytes treated with control siRNA or siRNA targeting *C3orf18*. All siRNA knockdowns were performed post-differentiation, with the first siRNA transfection on day 6, and again on day 10. Experiments were performed on day 14 after differentiation.

Therefore, the discrepancy identified by the Reviewer is explained by the “day 0” and “siControl” conditions in 4a and 4b being distinct. In 4a, the day 0 is undifferentiated cells, and the siControl in 4b is differentiated adipocytes subjected to non-targeting siRNA.

We have updated the legend for Extended Data Fig. 4b:

Western blot analysis of C3ORF18 expression in 3T3-L1 adipocytes with siRNA-mediated depletion of C3ORF18. Knockdown was performed on day 6 and day 10 of differentiation

and cells were harvested on day 14 (8 d knockdown). 10% SDS-PAGE gel used to resolve proteins, full membrane shown on right.

Additionally, the nature of the apparent MW shift observed for C3ORF18 should be addressed. What post-translational modifications (e.g., glycosylation) could account for this substantial shift on SDS-PAGE? It would be useful to treat the samples with glycosidases to determine whether the migration pattern changes, supporting the presence of glycosylation or other modifications.

We thank the reviewer for their comment. We have been puzzled by this observation since starting our work on this protein. We now include additional data in the manuscript to provide more detailed analysis of C3ORF18 post-translational modifications and to try and shed light on the molecular weight shift.

1. N-linked glycosylation likely contributes to the molecular weight shift since C3ORF18 has a canonical N-link glycosylation motif in its luminal N-terminal domain. Indeed, treating lysates from HEK-293 cells overexpressing HA-C3ORF18 with PNGaseF to remove all N-linked glycosylation results in an approximate 2-3 kDa decrease in the apparent C3ORF18 molecular weight. Mutation for Asn47 to Ala had the same effect of the apparent molecular weight, and this mutant was insensitive to PNGaseF. These data indicate that C3ORF18 is glycosylated at Asn47, but this only partly explains the higher-than-expected molecular weight on SDS-PAGE, suggesting that additional biochemical properties of C3ORF18 must underlie the major mobility shift.
2. Proteins can also be glycosylated on Ser/Thr residues. To test for O-linked glycosylation, we used a kit-based method to remove O-glycans, which relies on β -elimination, a base-induced chemical cleavage of the O-glycosidic bond that frees the sugar from Ser and Thr residues while preserving both the protein and glycan. This treatment did not affect the apparent molecular weight of C3ORF18 on SDS-PAGE.
3. Lysine residues can be modified in several ways, including the addition of small proteins like ubiquitin and SUMO that can substantially alter protein MW. To test whether lysine modification may explain increased C3ORF18 molecular weight, we generated a K-null mutant. Removing all lysines did not affect the apparent molecular weight of C3ORF18 on SDS-PAGE.
4. Phosphorylation can lead to large changes in electrophoretic mobility. However, C3ORF18 is not reported to be phosphorylated on phosphositeplus. Accordingly, dephosphorylation of protein lysates from cells expressing HA-C3ORF18 using lambda phosphatase did not affect the apparent molecular weight of C3ORF18 on SDS-PAGE.

5. Since the apparent weight of C3ORF18 appeared somewhat consistent with a dimer ($17 + 17 = 34$ kDa), we tested whether C3ORF18 homodimerises. We expressed two differently tagged C3ORF18 fusions in HEK cells (HA-C3ORF18; Alfa-C3ORF18) or a C3ORF18 fusion with both epitope tags (HA-C3ORF18-Alfa) and immunoprecipitated transiently expressed C3ORF18 using an anti-HA antibody. We did not detect an anti-Alfa signal in the precipitate, unless the HA and Alfa tags were on the same C3ORF18 molecule (i.e., dual tagged). These data suggest C3ORF18 does not homodimerise.
6. Finally, we tested whether intrinsically disordered regions in C3ORF18 may cause decreased migration by SDS-PAGE, as described for other proteins (PMID: 11420437). The N-terminal region of C3ORF18 is highly disordered as seen on IUPRED2a. Removal of the N-terminus almost completely restored C3ORF18 SDS-PAGE migration to its expected molecular weight. These data suggest that a feature of the luminal N-terminal region of C3ORF18, perhaps the disordered regions, which are enriched in hydrophilic and charged residues that bind SDS poorly, is responsible for decreased electrophoretic mobility on SDS-PAGE.

These data provide additional information on C3ORF18 assessment by Western blot and validation that the band observed at ~40 kDa is C3ORF18. are in **New Extended Data Figure 4d-g**.

Text on page 6:

Additional biochemical studies revealed that the N-terminal luminal portion of C3ORF18 was responsible for its reduced electrophoretic mobility, perhaps because of high disorder in this region⁵⁹ (Extended Data Fig. 4d-g).

Extended data figure legends:

d) Intrinsic disorder prediction of C3ORF18 (upper panel) generated using the IUPred3A server (iupred.elte.hu)⁹². Y-axis represents the disorder probability score, and x-axis shows amino acid position. The grey horizontal line (0.5) indicates the disorder threshold, with regions above this value predicted to be intrinsically disordered. HA-C3ORF18 with indicated amino acid residues deleted (lower panel) revealed that the disordered N-terminus is responsible for most of the apparent molecular weight increase observed by SDS-PAGE. e) Deletion of all lysine (K) residues did not alter the apparent molecular weight of C3ORF18, ruling out lysine modifications as the cause of the molecular weight shift. f) PNGase treatment, and mutation of Asn47 to Ala (N47A) increased C3ORF18 migration by SDS-PAGE, indicative of N-linked glycosylation at Asn47. The shift in the apparent molecular weight after N-glycan removal suggests that N-linked glycosylation contributes, albeit marginally, to the higher-than-expected observed molecular weight for C3ORF18. g) O-glycan removal did not alter the apparent molecular weight of C3ORF18.

To strengthen Figure 4b, I suggest including DAPI staining to help assess whether C3ORF18 localizes to the nucleus or perinuclear region. In the current images, the distinction is unclear, and co-staining would provide valuable spatial context. Moreover, in basal conditions, the apparent association of C3ORF18 with ribosomes (Figure 3d) suggests a possible localization to the rough ER, which aligns with its predicted

membrane-associated nature. Colocalization experiments with ER markers, or proximity ligation assays (PLA), would help confirm this interpretation.

To address the Reviewers comments, we have now performed more extensive colocalisation analysis at higher resolution for C3ORF18 using a series of subcellular markers (EEA1, TGN46, Calnexin), TfR and GLUT4. These data are presented in **New Extended Data Figure 4i**.

C3ORF18 does not colocalise with calnexin, implying it is not ER-localised. The perinuclear staining of C3ORF18 is juxtaposed to TGN46 (as observed for TfR and G4), suggesting this is a post-Golgi compartment. C3ORF18 had limited overlap with the early endosome marker EEA1, but a high degree of colocalisation with TfR in both the perinuclear region and periphery. Comparatively, the overlap with GLUT4 in these same regions was lower. Importantly, this analysis also identified that C3ORF18 can be found in vesicles that are both TfR- and GLUT4-negative. This suggests that C3ORF18 may reside, at least partially, in a distinct vesicle population to both TfR and GLUT4. We have updated the text to describe these new data and to clarify the degree of C3ORF18 colocalisation with TfR and GLUT4.

See page 6 of the text:

To assess C3ORF18 subcellular localisation and validate the insulin-responsive relocalisation observed in our proteomics (Fig. 3c-e) and Western blot data (Extended Data Fig. 4c), we immunostained endogenous C3ORF18 in adipocytes under basal and insulin-stimulated conditions. This revealed C3ORF18 localised to both the perinuclear region and cell periphery, with increased peripheral staining in response to insulin (Fig. 4c-d). We observed similar localisation in adipocytes over-expressing HA-C3ORF18 (Extended Data Fig. 4h). HA-C3ORF18 did not colocalise with the ER-marker calnexin, and the perinuclear staining of HA-C3ORF18 was juxtaposed to the trans-Golgi marker TGN46 (Extended Data Fig. 4i), suggesting this is a post-Golgi compartment. HA-C3ORF18 had limited overlap with the early endosome marker EEA1 (Extended Data Fig. 4i), but endogenous C3ORF18 substantially colocalised with TfR and GLUT4, particularly in the perinuclear region (Fig. 4d). Quantitative analysis revealed greater colocalisation of C3ORF18 with TfR than GLUT4 (Fig. 4e), best exemplified when assessing C3ORF18 and TfR immunostaining in punctate structures towards the cell periphery under basal conditions (Fig. 4d). Importantly, this analysis also identified that C3ORF18 can be found in vesicles that are both TfR- and GLUT4-negative (Fig. 4d, inset). This suggests that C3ORF18 may reside, at least partially, in a distinct vesicle population to both TfR and GLUT4.

The observed relocalization of C3ORF18 following inhibition of PI3K and AKT appears robust; however, a representative image would help the reader better appreciate the morphological differences under these conditions.

We thank the reviewer for their suggestion. We have added images to complement the quantitative data in **New Figure 4g**.

As an additional analysis, I recommend exploring C3ORF18 dependency using large-scale CRISPR screen datasets, filtered for adipocyte or metabolic cell types. This could offer insights into functional interactions or pathways related to C3ORF18.

We thank the reviewer for their suggestion. We have looked into these datasets, specifically on the Welcome to the BioGRID Open Repository of CRISPR Screens (ORCS). The most biologically relevant cell type in which screening had been conducted is the myoblast line C2C12, where *c3orf18* knockout resulted in accelerated mitophagy, implicating C3ORF18 as a negative regulator of this process (PMID: 31618756). However, a wide range of proteins involved in membrane trafficking were also identified in this screen (e.g., ESCRT proteins), so mitophagy may not represent the primary role for C3ORF18. We thank the reviewer for pointing us towards this information, but have decided to include this in our future studies and not refer to this in the current manuscript.

Lastly, the term "PM" (plasma membrane) is used in the abstract without being defined. Please define it upon first use.

Thank you for highlighting this omission. We have updated the abstract.

Reviewer #3 (Remarks to the Author):

In their study, Conway et al. use LOPIT, a subcellular proteomics approach, in a 3T3-L1 mouse cell model to screen for protein relocalizations following insulin stimulation. They identify C3ORF18, a protein of unknown function, as a potential novel player in insulin signaling. The manuscript is clearly structured, and the data are clearly presented and accessible via a Shiny app. The approach is interesting and allows for the unbiased identification of novel components involved in signaling.

However, the biological relevance of the study is somewhat limited, and the overall outcome modest. The new insights gained are relatively minor, and the follow-up analysis of C3ORF18 is very superficial. The function of the protein remains unclear, and its role in insulin signaling is not convincingly established. In addition, the information about the used methods in the text itself is too short and difficult to follow, lacking the necessary detail to fully understand the data.

Overall, while the technical approach is of interest, the study would benefit from a more thorough characterization of C3ORF18 and clearer methodological documentation. As it stands, the manuscript presents only limited biological insight.

Specific points:

Please provide a clearer explanation in the main text of how the LOPIT approach works, including the principle of fractionation, and how BUNDLE assigns proteins to compartments based on distribution profiles. It is currently unclear how marker sets are chosen and whether they are assumed to remain static between conditions.

We thank the reviewer for their comment and acknowledge greater explanation of the LOPIT-DC workflow and analysis is beneficial to reader interpretation.

We have included more experimental and analytical details on page 3 of the text, in the sections entitled: ***Mapping the subcellular 3T3-L1 adipocyte proteome using LOPIT-DC and Insulin stimulates widespread protein relocalisation.***

- The number of markers used appears large, and it seems they are not reassigned under insulin stimulation. Does the method assume these markers are static and do not relocalize? If so, this is a potential limitation.

The method assumes that marker proteins do not change condition under stimulation. This is not a limitation of the approach but a necessary modelling constraint. Marker proteins, by definition, are used to define a reference frame for subcellular niches in each condition, rather than as targets of differential localisation. Requiring that markers do not re-localise ensures identifiability of the underlying Bayesian mixture components and meaningful comparison of subcellular compartments across conditions.

The number of markers used in this study is comparable to, and often less than that used in previously published subcellular spatial proteomics (PMID: 30659192, 39617061, 34599159, 38565923) and serves to improve robustness and reduce uncertainty in protein assignments. This assumption applies only to the markers and does not restrict the ability of the method to detect widespread or condition-specific relocalisation among non-marker proteins.

We have added to our methods section to further explain how marker proteins were selected and used:

The data was annotated with marker proteins, i.e. proteins well established to localise to a specific subcellular compartment, from previous subcellular proteomics experiments^{12,15,77}. Additional marker proteins were identified through examining protein localisation in UniProt for proteins reported only to be localised to one organelle, and by mining the literature. Proteins were manually excluded as marker proteins if they appeared to relocalise in

response to insulin. Under de Duve's principle⁷⁸, proteins residing in the same subcellular compartment are expected to share characteristic fractionation profiles. BUNDLE leverages this by comparing the profiles of proteins with unknown localisation to those of curated organelle markers to infer subcellular assignments.

- While some organelle profiles (e.g., mitochondria) are clearly separated, others, such as plasma membrane and Golgi, appear overlapping and are harder to distinguish. This is also reflected in their lower concordance with published data. Please provide a quantitative metric or quality score for the compartment assignments to support interpretability.

We thank the reviewer for their comment. To quantitatively assess whether our dataset achieved sufficient separation of subcellular compartments we applied the QSep metric (PMID: 3071172) from the R pRoloc package to determine the separation of organelles in our subcellular proteomics datasets. The raw QSep score, represents the average Euclidean distance between clusters and therefore gives a direct measure of how far apart different compartments lie. QSep scores are then normalised, which refines this measure by taking into account the internal compactness of the reference cluster; it is calculated as the ratio of the average between-cluster distance to the average within-cluster distance. Thus, the normalised score reflects not only the separation of clusters but also how tightly each one is defined. For both metrics, larger scores indicate better resolution of compartments. We have added a boxplot summarising the distribution of normalised QSep values across all cluster pairs under both conditions in **New Extended Data Figure 1a** From the QSep scores, we conclude that most compartments are generally well-resolved in our dataset, with the nucleus, proteasome and, the cytosol being the best-resolved compartments.

This plot is included in Extended Data Fig 1a, and are described in the manuscript in the following sections:

Results - QSep, which quantifies the separation of clusters in subcellular proteomics data, was used to confirm the separation of organelles in our annotated data (Extended Data Fig. 1a).

Methods – QSep calculates the Euclidean distance between raw protein profiles for each set of organelle markers, with higher values indicating better separation in a multi-dimensional space.

For completeness, below, we provide the reviewers with all raw and normalised QSep values under both basal and insulin-stimulated conditions.

Reviewer Figure 3: Heatmaps showing raw (left) and normalised (right) QSep scores under basal (top) and insulin-stimulated (bottom) conditions. The diagonal represents within-cluster distances (compactness), whereas off-diagonal values represent separation between different compartments. Colours range from blue (short distances) to red (large distances), with higher values indicating greater separation.

- Line 111: If proteins in the ER–Golgi–PM axis show low concordance with other datasets, how can changes in localization be confidently attributed to insulin signaling rather than inherent trafficking variability?

We thank the reviewer for their comment and understand the Reviewers concern. When referring to ‘low concordance’, we mean only in terms of allocation of localisation, not with regard to movement. It is true that proteins within the ER-Golgi-PM pathway are harder to resolve using the differentiation centrifugation method we have used here. We also know that proteins are highly mobile between these compartments, and that they may multilocalise. As such, we agree that mapping of proteins to these organelles is challenging.

However, we are confident in attributing changes in localisation to the insulin stimulus because changes in localisation of a protein in response to insulin were not dependent

on allocation of protein to an organelle. Protein movement was predicted in BUNDLE based on individual protein profiles, and did not rely on information on protein localisation.

We have clarified that the process of identifying protein relocalisation was independent of protein allocation to organelles in the text on page 4:

“This Bayesian analysis framework compares the distribution profiles of proteins in unstimulated (basal) and insulin-stimulated adipocytes, and is independent of whether a protein has been allocated to an organelle (as in Fig. 1e).”

- From ~4000 detected proteins, only ~1700 were assigned to compartments. Please explain why the remaining proteins could not be assigned. What thresholds or criteria (e.g., probability cutoffs) were used and how was it set?

All proteins in BUNDLE receive a localisation assignment, which is accompanied by several probabilistic measures, two of which are:

- Allocation probability – the mean posterior probability that a protein belongs to a specific subcellular niche (i.e. how likely the protein is to reside in that organelle).
- Outlier probability – the posterior probability that a protein belongs to the outlier component rather than any annotated organelle (i.e. the probability that the protein is not well described by any of the annotated subcellular niches).

To assign proteins confidently to a single organelle, we filter based on both measures. Allocation probabilities closer to 1 indicate high confidence, while outlier probabilities are inverted during processing (1 - outlier), so values closer to 1 indicate stronger likelihood of being an outlier. As detailed in the Methods, proteins were assigned to a subcellular niche if their allocation probability was >0.9 and their outlier probability was below the median outlier probability for that organelle. This ensures that only proteins with high-confidence, single-organelle localisation are assigned, avoiding 'false-positive' assignments. Full annotated code for these steps is publicly available alongside the manuscript at https://github.com/CambridgeCentreForProteomics/adipocyteLOPIT2025/blob/main/4_processing_bundle/4_processing_bundle.Rmd.

That only ~1,700 of ~4,000 detected proteins are assigned reflects both the conservative nature of probabilistic thresholding and biological reality. Many proteins reside in multiple subcellular locations (PMID: 32321741) or exhibit context-dependent localisation and are therefore not represented well by a single high-confidence assignment. Rather than being a limitation, this thresholding is deliberate: it avoids overconfident or misleading classifications, while still retaining the underlying

probabilities for all proteins. The posterior probability for every protein is available in the Supporting Information Table 1 and the accompanying Shiny App <https://proteome.shinyapps.io/adipocyte2025/>, allowing readers to interrogate uncertain or multi-localised proteins quantitatively.

Are proteins with lower assignment confidence also more likely to show differential localization?

A majority of proteins predicted to be differentially localised were not assigned to an organelle in at least one condition. This is intuitive; many proteins redistribute partially between compartments rather than moving entirely from one organelle to another. Consequently, proteins with lower-confidence single-organelle assignments are indeed more likely to show differential localisation, reflecting both the biology and the probabilistic framework of the method.

- Please describe in more detail in the text what the second proteomics method is. Can you compare the changes in the PM detected via LOPIT-DC and biotinylating? What is different and why? Why was it necessary in addition to LOPIT? What new information does it provide that LOPIT does not?

We thank the reviewer for their comment.

The LOPIT-DC technique identifies protein localisation and/or movement by comparing proteins distribution profiles across different obtained by differential centrifugation. This method is very accurate in localising proteins to organelles that have a very distinct profile across fractions, but less able to map proteins to less well resolved organelles (e.g., ER-Golgi-PM). Further, while it detects protein movement through altered protein profiles, these data are not quantitative and also often do not provide clear information on where proteins are moving to and from. Since we are particularly interested in proteins that are delivered to the PM in response to insulin (e.g., GLUT4), we undertook this complementary PM-specific analysis, which has been used extensively to profile changes in the PM proteome (PMID:22292497; 27462310; 32165698; 37302069). This provides 1) additional high-confidence assignment of proteins to the PM and 2) more quantitative data on the extent of changes in protein abundance at the PM.

Using these two orthologous proteomics approaches concurrently enabled us to identify proteins which were both predicted to move in LOPIT-DC and significantly change in abundance at the PM, giving us greater confidence these proteins underwent insulin-stimulated translocation to the PM.

We have included additional justification for and description of this technique in our revised manuscript (page 5):

LOPIT-DC revealed the adipocyte PM proteome to be highly insulin responsive (Fig 3a, Extended Data Fig. 3a). To provide a more quantitative assessment of protein abundance

changes at the PM in response to insulin, we used an orthologous cell surface biotinylation proteomics approach¹³ (Fig. 3b). Briefly, 3T3-L1 adipocytes were left untreated or stimulated with insulin (100 nM, 20 min) before surface proteins were biotinylated and isolated by affinity purification.

- What happens to lipid droplets in the P1–P8 fractions? Could LD-associated proteins be counted more than once due to their presence in both the LD fraction and earlier fractions?

We thank the reviewer for their comment. Whilst every care was taken to remove lipid droplets from fraction 1-8 and the cytosol, it is possible some LDs remain. This is why the abundance of LD proteins is not 0 in the earlier fractions. Further, as LDs bud from the ER, some LD-associated proteins are likely to be genuinely present in the non-LD fractions (i.e., not contaminants). Nevertheless, as shown in Extended Data Fig. 1b and 1c, classical LD proteins, including perilipins, are clearly enriched in the LD fraction.

Due to the way that data from both centrifugation fractions and isolated LDs were combined and analysed ensures that these are not counted 'twice'. The 10 fractions of each sample (P1-8, cytosol and LD) are each labelled with a TMT-tag and combined into one sample. Therefore, the amount of a LD protein across all fractions is quantified, and normalised to give its relative distribution across all fractions.

- Septins are discussed in the context of lipid droplets. Do any septins relocate in response to insulin? What is their possible role in adipocyte biology and lipid storage?

We thank the reviewer for their comment. In our data, Septins were found to be present in the lipid droplet fraction in the basal state, but did not relocate with insulin. Septin 11 has previously been implicated in adipogenesis by regulating neutral lipid loading onto lipid droplets (PMID: 27866222). We have now confirmed punctate localisation of both Septin 2 and Septin 11 on the LD surface in both murine and human adipocytes (**New Figure 1f and Extended Data Figure 1h**). Further, *Sept2* knockdown during 3T3-L1 differentiation has revealed a role for Septin 2 in lipid droplet accumulation during differentiation, despite no obvious changes in adipogenic markers (**New figure 1g and Extended Data Figure 1l-k**). While our observations further lend weight to a role for Septins in lipid accumulation and lipid droplet biology, they do not provide additional mechanistic insight than already provided for Sept7, 9 and 11 (PMID: 40015624; 27866222; 35573204; 27417143). Further mechanistic studies will be required to understand the exact biology behind this process.

- Interpretation of ER accumulation upon insulin: Some proteins show ER accumulation upon insulin treatment. How do you interpret this? Could this indicate a block or re-routing in vesicular trafficking rather than a classical translocation?

We thank the reviewer for their comment. Indeed, this relocalisation, largely from the PM to the ER, could represent a slowing of traffic through the secretory system as suggested, although we might not expect to detect this signal after only 30 min insulin stimulation. Further work would be needed to ascertain the validity and mechanisms of these putative PM-to-ER translocations.

- Protein abundance vs. localization change (Line 138): For IRAP, the differential localization probability is high, but the shift appears to reflect abundance rather than spatial redistribution. How does your model distinguish these scenarios?

We thank the reviewer for their comment. The profile plots to display protein abundance across LOPIT fractions are normalised so that the total signal across all fractions = 1 for both basal and insulin samples. In correlation profiling methods, we are interested in the normalised distribution pattern (i.e. relative normalised abundance), rather than raw abundances typically use as input for statistical analysis in a classic total proteome experiment. As such, it is not possible to observe changes in protein abundance from these plots.

The Reviewer is correct that it is important to consider how changes in abundance and localisation interact over longer intervention periods. There is no change in total cellular IRAP or GLUT4- we have previously shown that there are no changes in protein abundance across the proteome with 30 min insulin stimulation (PMID: 36808134). We have included these data for IRAP and GLUT4 in **New Extended Data Figure 2a**.

See page 4: Known insulin-responsive proteins including GLUT4⁴⁹ and LNPEP/IRAP^{50,51} were predicted to move with high-confidence (*diff. loc. prob.* = 1, Fig. 2b), validating our approach to detect insulin-responsive protein translocation. Changes in protein localisation were independent of IRAP and GLUT4 protein abundance (Extended Data Fig. 2a), and we have previously reported that total protein abundance is unchanged across the proteome in response to acute insulin stimulation¹⁰.

- Line 146: What is the exact *diff. loc. prob.* threshold used to define significant relocalization? For proteins like mTOR, the localization shift is not obvious from the profiles—please clarify.

We thank the reviewer for their comment. The *diff. loc. prob.* threshold's used to define relocalisation is described in the Methods and Results sections, as follows:

Methods - Proteins were assigned to a subcellular niche if their allocation probability was >0.9 and their outlier probability was < median outlier probability for that organelle. Proteins were deemed differentially localised if differential localisation probability > 0. Proteins with differential localisation probability = 1 were considered 'high confidence' and all others as 'candidates'

Results - We identified 899 out of 4005 proteins as candidate movers (differential localisation probability (*diff. loc. prob.*) > 0), of which 502 were high-confidence (*diff. loc. prob.* = 1) (Fig. 2a; Supplementary Table 1).

The reviewer is right that some profiles, like mTOR, show more subtle responses. In this case the differential localisation probability of mTOR was 0.14, indicating this is a lower confidence ('candidate') mover. Nevertheless, there is a clear distinction between the profiles in fraction 4, in which the lysosome markers were enriched (Extended Data Fig. 2c).

- Line 153: When proteins like DEPDC5 or SZT2 are said to dissociate from lysosomes, where do they relocalize? Is this to the cytosol or another compartment?

We thank the reviewer for their comment. Unfortunately, as these proteins were not assigned to an organelle with high confidence, we are unable to confidently say where they relocalised to under insulin-stimulated conditions. The profile of these proteins show they are more abundant in fractions 6,7 and 8 in the insulin-treated cells. As marker proteins for the ribosomes, proteasome and Golgi are all enriched in these fractions, it is possible these proteins are relocalising to one or more of these organelles. There is no increase in relative abundance of these proteins in fraction 9, the cytosolic fraction, so we cannot claim they relocalise to the cytosol. However, additional approaches (i.e., immunofluorescence) would be required to identify where these proteins relocalise to in response to insulin.

- Line 195: Has C3ORF18 been identified in other proteomic studies of adipocyte differentiation? If so, please cite relevant references.

C3ORF18 was identified by Klingelhuber and colleagues (PMID: 38565923) in SGBS preadipocytes and adipocytes, and differentiated human adipocyte precursor cells in subcellular proteomic studies, but was not quantified in their studies on the adipocyte proteome during differentiation. We were unable to find other studies that had identified C3ORF18 in their analyses (e.g., PMID: 32384580; 32991178).

- Figures 4b and 4e: The nuclear localization of C3ORF18 is suggested, but no nuclear marker is shown. Likewise, endosomal markers should be used to support the claim that the protein localizes to insulin-responsive endosomes, and

- Clarify C3ORF18 relocalization dynamics: The current description is ambiguous. Does C3ORF18 move from the nucleus, endosomes, or another compartment to the plasma membrane? How do endosomes fit into the proposed mechanism?

We thank the reviewer for their comments. We suggest C3ORF18 localises to the perinuclear region (around the nucleus), not within the nucleus.

We have now performed more extensive colocalisation analysis at higher resolution for C3ORF18 using a series of subcellular markers (EEA1, TGN46, Calnexin), TfR and GLUT4. These data are presented in **New Extended Data Figure 4i**.

C3ORF18 does not colocalise with calnexin, implying it is not ER-localised. The perinuclear staining of C3ORF18 is juxtaposed to TGN46 (as observed for TfR and G4), suggesting this is a post-Golgi compartment. C3ORF18 had limited overlap with the early endosome marker EEA1, but a high degree of colocalisation with TfR in both the perinuclear region and periphery. Comparatively, the overlap with GLUT4 in these same regions was lower.

The colocalisation of C3ORF18 with peripheral TfR vesicles provides evidence that it localises to endosomes. As TfR is insulin-responsive, with similar insulin response and translocation kinetics to C3ORF18, we suggest that C3ORF18 may translocate to the PM with TfR from this endosomal compartment.

See page 6 of the text:

To assess C3ORF18 subcellular localisation and validate the insulin-responsive relocalisation observed in our proteomics (Fig. 3c-e) and Western blot data (Extended Data Fig. 4c), we immunostained endogenous C3ORF18 in adipocytes under basal and insulin-stimulated conditions. This revealed C3ORF18 localised to both the perinuclear region and cell periphery, with increased peripheral staining in response to insulin (Fig. 4c-d). We observed similar localisation in adipocytes over-expressing HA-C3ORF18 (Extended Data Fig. 4h). HA-C3ORF18 did not colocalise with the ER-marker calnexin, and the perinuclear staining of HA-C3ORF18 was juxtaposed to the trans-Golgi marker TGN46 (Extended Data Fig. 4i), suggesting this is a post-Golgi compartment. HA-C3ORF18 had limited overlap with the early endosome marker EEA1 (Extended Data Fig. 4i), but endogenous C3ORF18 substantially colocalised with TfR and GLUT4, particularly in the perinuclear region (Fig. 4d). Quantitative analysis revealed greater colocalisation of C3ORF18 with TfR than GLUT4 (Fig. 4e), best exemplified when assessing C3ORF18 and TfR immunostaining in punctate structures towards the cell periphery under basal conditions (Fig. 4d). Importantly, this analysis also identified that C3ORF18 can be found in vesicles that are both TfR- and GLUT4-negative (Fig. 4d, inset). This suggests that C3ORF18 may reside, at least partially, in a distinct vesicle population to both TfR and GLUT4.

- Line 242: The effect of C3ORF18 knockdown on AKT signaling is claimed, but no clear quantification is shown, and the Western blot quality is suboptimal. Please provide quantitative data and clearer blots.

We thank the reviewer for their comments. Quantification was shown in Extended Data Fig. 5c (**now in New Figure 5g and Extended Data Figure 5j**) and all blots are shown in the Source Data submitted alongside the revised manuscript.

- Predicted interactors of C3ORF18: Many predicted interactors are mitochondrial proteins. How does this reconcile with its observed localization and proposed function in insulin signaling?

We thank the reviewer for their comment. Both Biogrid (PMID: 33961781; HEK293 cells) and String (PMID: 36370105; *Saccharomyces cerevisiae*) databases include experimentally determined interactions between C3ORF18 and a subset of mitochondrial proteins. Firstly, these models may not recapitulate 1) endogenous C3ORF18 interactors as it is lowly expressed in HEK cells and 2) *Saccharomyces cerevisiae* may not recapitulate spatial organisation/interactome of a mammalian cell. Based on our current thinking about C3ORF18 function, we cannot currently explain these interactions. However, we note in our response to Reviewer 2, that C3ORF18 has been implicated in mitophagy in CRISPR screening, which may provide a link between C3ORF18 and mitochondria. Together, these observations will inform our future studies on C3ORF18 function.

We do not think that these interactions are strong evidence of mitochondrial localisation of C3ORF18. Mitochondria are one of the best resolved organelles within LOPIT, and indeed subcellular proteomics methods in general, as there are a large number of mitochondrial-specific proteins in mammalian cells which can be used to train the localization classification algorithm. In our data 256 proteins were used as mitochondrial marker proteins, more than any other organelle (Supplementary Table 1). C3ORF18 was also identified in our plasma membrane proteomics where surface proteins were labelled with biotin before cells were lysed, further supporting C3ORF18 as a protein recycling with the PM and not localised to mitochondria. Additionally, immunofluorescence staining of C3ORF18 does not resemble the typical morphology seen for mitochondria in adipocytes. Therefore, we are confident C3ORF18 is not localised to mitochondria.

- Line 248: The statement that C3ORF18 improves insulin sensitivity in adipose tissue is not supported by the presented data. This conclusion should be supported with additional experiments.

We thank the reviewer for their comment. Our intention was to highlight that the data obtained from cell work and these clinical correlations were directionally consistent. We have edited out statement for accuracy:

“These clinical correlative data are consistent with our data from cultured cells, indicating that C3ORF18 positively regulates insulin sensitivity in adipocytes.”

- The functional role of C3ORF18 doesn't become clear. Is its relocalization upon insulin treatment dependent on phosphorylation? Is its role primarily in signaling or trafficking? Based on your data, how might C3ORF18 mechanistically influence the insulin signaling cascade? A speculative model or hypothesis would strengthen the discussion.

We do not yet have a complete picture of the function of C3ORF18. Its relocalisation to the plasma membrane is dependent on insulin signalling via AKT, but C3ORF18 itself has not been found to be phosphorylated in response to insulin. We do not know the exact signalling and trafficking mechanisms that underpin the C3ORF18 translocation responses.

We have including additional text in our discussion to highlight the need for additional work here on Page 9.

Future studies will need to address the exact mechanisms underpinning insulin-stimulated C3ORF18 redistribution and C3ORF18 linking C3ORF18 and endosomal traffic to longer-term adipocyte insulin sensitivity, and the role C3ORF18 plays in other tissues where it is enriched, such as muscle.

We have provided additional data to try and link C3ORF18 and insulin signalling. Specifically, we have assessed the mRNA expression and protein abundance of key regulators of the insulin response in adipocytes upon *C3orf18* knockdown in differentiated adipocytes. We have also performed additional functional assays after 4 d knockdown to try and reveal the primary effect of loss of *c3orf18*.

1. After 8 days KD, we observed impaired insulin-stimulated GLUT4 and TfR translocation, impaired insulin-stimulated glucose transport and impaired insulin signalling to AKT and TBC1D4.
2. As Reviewer 1 points out, in the case of GLUT4, TfR and 2DOG uptake, we observed some effects under of *C3orf18* loss under basal conditions. Accordingly, the protein abundance of TfR, GLUT1 and GLUT4 was decreased by *C3orf18* knockdown. For TfR, mRNA expression was also decreased.
3. To focus on insulin signalling, we measured the abundance and localisation of insulin receptor (INSR) and IRS1. *C3orf18* knockdown did not affect INSR localisation, protein abundance or mRNA expression. However, there was a marked decrease in IRS1 protein.
4. In addition, *C3orf18* knockdown decreased adipocyte marker gene mRNA expression when depleted from mature adipocytes.
5. In light of these wide-ranging effects, including transcriptional changes, after 8 days of *C3orf18* knockdown, we conducted limited experiments after 4 days knockdown, when C3ORF18 protein remained at ~60%. At this timepoint we detected no change in insulin signalling or GLUT4 translocation or abundance. However, surface and total TfR abundance were decreased. These data suggest that TfR is more sensitive to loss of C3ORF18.

These data are now all included in **New Figure 5 and Extended Data Figure 5**. Together, these data suggest that a primary effect of loss of C3ORF18 is altered TfR trafficking and/or stability, or at least that this pathway is more sensitive to C3ORF18 loss.

Prolonged or more exaggerated loss of C3ORF18 decreased GLUT1 and GLUT4 protein

in the absence of mRNA changes, suggesting increased protein turnover, and decreased insulin signalling responses, likely via lower IRS1 abundance. The combined effects of these changes in protein abundance impaired insulin-stimulated glucose transport.

We have outlined our current proposed model for the role that C3ORF18 plays in maintaining adipocyte insulin sensitivity in the manuscript text in page 9:

The molecular mechanisms linking C3ORF18 to adipocyte insulin responses remain unclear, although our data from cells depleted of *C3orf18* for shorter periods suggest the primary effect of loss of C3ORF18 is an impairment in endosomal trafficking, as indicated by loss of TfR protein and impaired TfR delivery to the PM. Our working model is that C3ORF18 promotes proper endosomal recycling, and impairing this process over longer periods disrupts adipocyte insulin responses via a range of mechanisms, including 1) direct effects on membrane traffic; 2) loss of key insulin-responsive proteins that traffic via the endosomal system (i.e., GLUT4); and 3) transcriptional responses that suggest a role for C3ORF18 in maintaining mature adipocyte identity.

Minor comments:

- Line 43: phrasing should be improved.

We have edited this sentence to improve clarity.

“Insulin binding to its receptor, INSR, alters the subcellular distribution of specific proteins via several mechanisms. These include modulating protein-protein interactions via protein phosphorylation, protein recruitment to membranes through lipid messenger generation (e.g., PI(3,4,5)P3), and direct effects on membrane trafficking.”

- Line 123 and 126: Extended data Fig.1h is mentioned before Fig.1g.

We thank the reviewer for highlighting this error. The text has been corrected.

- Please change colours of PM markers (basal vs insulin) in Fig. 3e so those can be distinguished by the reader.

We thank the reviewer for their suggestion. The colours have been updated so the PM markers in basal are in pink, to reflect the colour for this organelle throughout the paper, and the PM markers in the insulin treated cells in blue, as used to show profiles following insulin treatment throughout the manuscript.

- Any idea on what the band at 80-90kDa is in the C3ORF18 blots?

We thank the reviewer for their question. From our experience with this antibody, including samples where we have knocked-down C3ORF18, we conclude this is non-specific band. We have now annotated this band in the revised manuscript figure (**Extended Data Figure 4b**).

· Line 204: there is no figure 3f.

We thank the reviewer for highlighting this error. The text has been updated to the correct figure.